# *Storchastic*: A Framework for General Stochastic Automatic Differentiation

**Emile van Krieken**
Vrije Universiteit Amsterdam
e.van.krieken@vu.nl

**Jakub M. Tomczak**
Vrije Universiteit Amsterdam
j.m.tomczak@vu.nl

**Annette ten Teije**
Vrije Universiteit Amsterdam
annette.ten.teije@vu.nl

## Abstract

Modelers use automatic differentiation (AD) of computation graphs to implement complex deep learning models without defining gradient computations. Stochastic AD extends AD to stochastic computation graphs with sampling steps, which arise when modelers handle the intractable expectations common in reinforcement learning and variational inference. However, current methods for stochastic AD are limited: They are either only applicable to continuous random variables and differentiable functions, or can only use simple but high variance score-function estimators. To overcome these limitations, we introduce *Storchastic*, a new framework for AD of stochastic computation graphs. *Storchastic* allows the modeler to choose from a wide variety of gradient estimation methods at each sampling step, to optimally reduce the variance of the gradient estimates. Furthermore, *Storchastic* is provably unbiased for estimation of any-order gradients, and generalizes variance reduction techniques to any-order derivative estimates. Finally, we implement *Storchastic* as a PyTorch library at github.com/HEmile/storchastic.

## 1 Introduction

One of the driving forces behind deep learning is automatic differentiation (AD) libraries of complex computation graphs. Deep learning modelers are relieved by accessible AD of the need to implement complex derivation expressions of the computation graph. However, modelers are currently limited in settings where the modeler uses intractable expectations over random variables [37, 8]. Two common examples are reinforcement learning methods using policy gradient optimization [49, 29, 36] and latent variable models, especially when inferred using amortized variational inference [34, 20, 41, 43]. Typically, modelers estimate these expectations using Monte Carlo methods, that is, sampling, and resort to gradient estimation techniques [37] to differentiate through the expectation.

A popular approach for stochastic AD is reparameterization [20], which is both unbiased and has low variance, but is limited to continuous random variables and differentiable functions. The other popular approach [49, 45, 13] analyzes the computation graph and then uses the score function estimator to create a *surrogate loss* that provides gradient estimates when differentiated. While this approach is more general as it can also be applied to discrete random variables and non-differentiable functions, naive applications of the score function will have high variance, which leads to unstable and slow convergence. Furthermore, this approach is often implemented incorrectly [13], which can introduce bias in gradients.

We therefore develop a new framework called *Storchastic* to support deep learning modelers. They can use *Storchastic* to focus on defining stochastic deep learning models without having to worry about complex gradient estimation implementations. *Storchastic* extends DiCE [13] to other gradient estimation techniques than basic applications of the score function. It defines a surrogate loss by decomposing gradient estimation methods into four components: The proposal distribution, weight-

35th Conference on Neural Information Processing Systems (NeurIPS 2021).

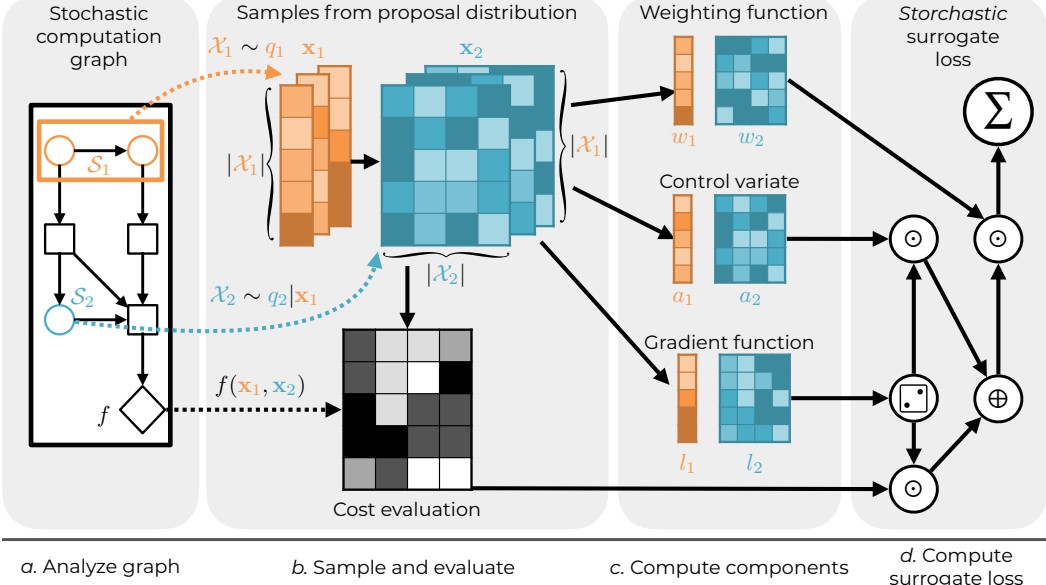

Stochastic computation graph · Samples from proposal distribution · Weighting function · *Storchastic* surrogate loss

$\mathcal{X}_1 \sim q_1$ · $\mathbf{x}_1$ · $\mathbf{x}_2$

$\mathcal{S}_1$

$|\mathcal{X}_1|$ · $|\mathcal{X}_1|$

$\mathcal{S}_2$ · $\mathcal{X}_2 \sim q_2 | \mathbf{x}_1$

$|\mathcal{X}_2|$

$w_1$ · $w_2$

Control variate · $a_1$ · $a_2$

Gradient function · $l_1$ · $l_2$

$f$ · $f(\mathbf{x}_1, \mathbf{x}_2)$

Cost evaluation

*a.* Analyze graph · *b.* Sample and evaluate · *c.* Compute components · *d.* Compute surrogate loss

Figure 1: An illustration of the (parallelized) *Storchastic* loss computation. *a.* Assign the stochastic nodes of the input stochastic computation graph (SCG) into two topologically sorted partitions. *b.* Evaluate the SCG. We first sample the set of values $\mathcal{X}_1$ from the proposal distribution. For each of the samples $\mathbf{x}_i \in \mathcal{X}_1$, we then sample a set of samples $\mathcal{X}_2$. The rows in the figure indicate different samples in $\mathcal{X}_1$, while the columns indicate samples in $\mathcal{X}_2$. The different samples are used to evaluate the cost function $f$ $|\mathcal{X}_1| \cdot |\mathcal{X}_2|$ times. *c.* Compute the weighting function, control variate and gradient function for all samples. *d.* Using those components and the cost function evaluation, compute the *storchastic* surrogate loss, mimicking Algorithm 1. $\odot$ refers to element-wise multiplication, $\oplus$ to element-wise summation and $\sum$ for summing the entries of a matrix.

ing function, gradient function and control variate. We can use this decomposition to get insight into how gradient estimators differ, and use them to further reduce variance by adapting components of different gradient estimators.

Our main contribution is a framework with a formalization and a proof that, if the components satisfy certain conditions, performing $n$-th order differentiation on the *Storchastic* surrogate loss gives unbiased estimates of the $n$-th order derivative of the stochastic computation graph. We show these conditions hold for a wide variety of gradient estimation methods for first order differentiation. For many score function-based methods like RELAX [16], MAPO [28] and the unordered set estimator [25], the conditions also hold for any-order differentiation. In *Storchastic*, we only have to prove these conditions locally. This means that modelers are free to choose the gradient estimation method that best suits each sampling step, while guaranteeing that the gradient remains unbiased. *Storchastic* is the first stochastic AD framework to incorporate the measure-valued derivative [40, 18, 37] and SPSA [46, 2], and the first to guarantee variance reduction of any-order derivative estimates through control variates.

In short, our contributions are the following:

1. We introduce *Storchastic*, a new framework for general stochastic AD that uses four gradient estimation components, in Section 3.1-3.3.

2. We prove Theorem 1, which provides conditions under which *Storchastic* gives unbiased any-order derivative estimates in Section 3.4. To this end, we introduce a mathematical formalization of forward-mode evaluation in AD libraries in Section 2.4.

3. We derive a technique for extending variance reduction using control variates to any-order derivative estimation in Section 3.5.

4. We implement *Storchastic* as an open source library for PyTorch, Section 3.7.

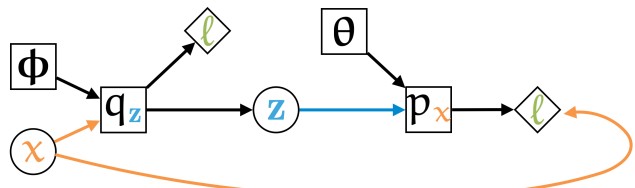

Figure 2: A Stochastic Computation Graph representing the computation of the losses of an VAE with a discrete latent space.

## 2 Background

We use capital letters $N, F, S_1, ..., S_k$ for nodes in a graph, calligraphic capital letters $\mathcal{S}, \mathcal{F}$ for sets and non-capital letters for concrete computable objects such as functions $f$ and values $\mathbf{x}_i$.

### 2.1 Stochastic Computation Graphs

We start by introducing Stochastic Computation Graphs (SCGs) [45], which is a formalism for stochastic AD. A *Stochastic Computation Graph* (SCG) is a directed acyclic graph (DAG) $\mathcal{G} = (\mathcal{N}, \mathcal{E})$ where nodes $\mathcal{N}$ are partitioned in *stochastic nodes* $\mathcal{S}$ and *deterministic nodes* $\mathcal{F}$. We define the set of *parameters* $\Theta \subseteq \mathcal{F}$ such that all $\theta \in \Theta$ have no incoming edges, and the set of *cost nodes* $\mathcal{C} \subseteq \mathcal{F}$ such that all $c \in \mathcal{C}$ have no outgoing edges.

The set of *parents* $\mathrm{pa}(N)$ is the set of incoming nodes of a node $N \in \mathcal{N}$, that is $\mathrm{pa}(N) = \{M \in \mathcal{N} | (M, N) \in \mathcal{E}\}$. Each *stochastic node* $S \in \mathcal{S}$ represents a random variable with *sample space* $\Omega_S$ and probability distribution $p_S$ conditioned on its parents. Each *deterministic node* $F$ represents a (deterministic) function $f_F$ of its parents.

$M$ *influences* $N$, denoted $M \prec N$, if there is a directed path from $M$ to $N$. We denote with $\mathcal{N}_{\prec N} = \{M \in \mathcal{N} | M \prec N\}$ the set of nodes that influence $N$. The *joint probability* of all random variables $\mathbf{x}_\mathcal{S} \in \prod_{S \in \mathcal{S}} \Omega_S$ is defined as $p(\mathbf{x}_\mathcal{S}) = \prod_{S \in \mathcal{S}} p_S(\mathbf{x}_S | \mathbf{x}_{\mathrm{pa}(S)})$, where $\mathbf{x}_{\mathrm{pa}(S)}$ is the set of values of the nodes $\mathrm{pa}(S)$. The *expected value* of a deterministic node $F \in \mathcal{F}$ is its expected value over sampling stochastic nodes that influence that node, that is,

$$\mathbb{E}[F] = \mathbb{E}_{\mathcal{S}_{\prec F}}[f_F(\mathrm{pa}(F))] = \int_{\Omega_{\mathcal{S}_{\prec F}}} p(\mathbf{x}_{\prec \mathcal{S}}) f_F(\mathbf{x}_{\mathrm{pa}(F)}) d\mathbf{x}_{\mathcal{S}_{\prec F}}.$$

### 2.2 Problem Statement

In this paper, we aim to define a *surrogate loss* that, when differentiated using an AD library, gives an unbiased estimate of the $n$-th order derivative of a parameter $\theta$ with respect to the expected total cost $\nabla_\theta^{(n)} \mathbb{E}[\sum_{C \in \mathcal{C}} C]$. This gradient can be written as $\sum_{C \in \mathcal{C}} \nabla_\theta^{(n)} \mathbb{E}[C]$, and we focus on estimating the gradient of a single cost node $\nabla_\theta^{(n)} \mathbb{E}[C]$.

### 2.3 Example: Discrete Variational Autoencoder

Next, we introduce a running example: A variational autoencoder (VAE) with a discrete latent space [20, 19]. First, we represent the model as an SCG: The *deterministic* nodes are $\mathcal{N} = \{\phi, \theta, q_z, p_x, \ell_{KLD}, \ell_{Rec}\}$ and the *stochastic* nodes are $\mathcal{S} = \{x, z\}$. These are connected as shown in Figure 2. The *parameters* are $\Theta = \{\theta, \phi\}$ which respectively are the parameters of the variational posterior $q$ and the model likelihood $p$, and the *cost nodes* $\mathcal{C} = \{\ell_{KLD}, \ell_{Rec}\}$ that represent the KL-divergence between the posterior and the prior, and the 'reconstruction loss', or the model log-likelihood after decoding the sample $z$. Finally, $q_z$ represents the parameters of the multivariate categorical distribution of the amortized variational posterior $q_\phi(z|x)$. This SCG represents the equation

$$\mathbb{E}_{x,z}[\ell_{KLD} + \ell_{Rec}] = \mathbb{E}_x[\ell_{KLD}] + \mathbb{E}_{x, z \sim q_\phi(z|x)}[\ell_{Rec}].$$

The problem we are interested in is estimating the gradients of these expectations with respect to the parameters. Since $x$ is not influenced by the parameters, we have $\nabla_\theta^{(n)} \mathbb{E}_x[\ell_{KLD}] = 0$

and $\nabla_\phi^{(n)}\mathbb{E}_x[\ell_{KLD}] = \mathbb{E}_x[\nabla_\phi^{(n)}\ell_{KLD}]$. The second term is more challenging. We can move the gradient with respect to $\theta$ in, since $z$ is not influenced by $\theta$: $\nabla_\theta^{(n)}\mathbb{E}_{x,z\sim q_\phi(z|x)}[\ell_{Rec}] = \mathbb{E}_{x,z\sim q_\phi(z|x)}[\nabla_\theta^{(n)}\ell_{Rec}]$. However, we cannot compute $\nabla_\phi^{(n)}\mathbb{E}_{x,z\sim q_\phi(z|x)}[\ell_{Rec}]$ without gradient estimation methods. This is because sampling from $q_\phi(z|x)$ is dependent on $\phi$. Furthermore, since we are dealing with a discrete stochastic node, we cannot apply the reparameterization method here without introducing bias.

## 2.4 Formalizing AD libraries and DiCE

To be able to properly formalize and prove the propositions in this paper, we introduce the 'forward-mode' operator that simulates forward-mode evaluation using AD libraries. This operator properly handles the common 'stop-grad' operator, which ensures that its argument is only evaluated during forward-mode evaluations of the computation graph. It is implemented in Tensorflow and Jax with the name `stop_gradient` [1, 4] and in PyTorch as `detach` or `no_grad` [39]. 'stop-grad' is necessary to define surrogate losses for gradient estimation, which is why it is essential to properly define it. For formal definitions of the following operators and proofs we refer the reader to Appendix A.

**Definition 1** (informal). The *stop-grad* operator $\perp$ is a function such that $\nabla_x \perp(x) = 0$. The *forward-mode* operator $\to$, which is denoted as an arrow above the argument it evaluates, acts as an identity function, except that $\overrightarrow{\perp(a)} = \overrightarrow{a}$. Additionally, we define the MagicBox operator as $\boxdot(x) = \exp(x - \perp(x))$.

Importantly, the definition of $\to$ implies that $\overrightarrow{\nabla_x f(x)}$ does not equal $\nabla_x \overrightarrow{f(x)}$ if $f$ contains a stop-grad operator. MagicBox, which was first introduced in [13], is particularly useful for creating surrogate losses that remain unbiased for any-order differentiation. It is defined such that $\overrightarrow{\boxdot(x)} = 1$ and $\nabla_x \boxdot(f(x)) = \boxdot(f(x))\nabla_x f(x)$. This allows injecting multiplicative factors to the computation graph only when computing gradients.

Making use of MagicBox, DiCE [13] is an estimator for automatic $n$th-order derivative estimation that defines a surrogate loss using the score function:

$$\nabla_\theta^{(n)}\mathbb{E}[\sum_{C\in\mathcal{C}} C] = \mathbb{E}\left[\overrightarrow{\nabla_\theta^{(n)}\sum_{C\in\mathcal{C}}\boxdot(\sum_{S\in\mathcal{S}_{\prec C}}\log p(\mathbf{x}_S|\mathbf{x}_{\text{pa}(S)}))C}\right]. \tag{1}$$

DiCE correctly handles the credit assignment problem: The score function is only applied to the stochastic nodes that influence a cost node. It also handles pathwise dependencies of the parameter through cost functions. However, it has high variance since it is based on a straightforward application of the score function.

## 3 The *Storchastic* Framework

In this section, we introduce *Storchastic*, a framework for general any-order gradient estimation in SCGs that gives modelers the freedom to choose a suitable gradient estimation method for each stochastic node. First, we present 5 requirements that we used to develop the framework in Section 3.1. *Storchastic* deconstructs gradient estimators into four components that we present in Section 3.2. We use these components to introduce the *Storchastic* surrogate loss in Section 3.3, and give conditions that need to hold for unbiased estimation in Section 3.4. In Section 3.5 we discuss variance reduction, in Section 3.6 we discuss several estimators that fit in *Storchastic*, and in Section 3.7 we discuss our PyTorch implementation. An overview of our approach is outlined in Figure 1.

### 3.1 Requirements of the *Storchastic* Framework

First, we introduce the set of requirements we used to develop *Storchastic*.

1. Modelers should be able to choose a different gradient estimation method for each stochastic node. This allows for choosing the method best suited for that stochastic node, or adding background knowledge in the estimator.

2. *Storchastic* should be flexible enough to allow implementing a wide range of reviewed gradient estimation methods, including score function-based methods with complex sampling techniques [52, 25, 28] or control variates [16, 47], and other methods such as measure-valued derivatives [18, 40] and SPSA [46] which are missing AD implementations [37].

3. *Storchastic* should define a *surrogate loss* [45], which gives gradients of the SCG when differentiated using an AD library. This makes it easier to implement gradient estimation methods as modelers get the computation of derivatives for free.

4. Differentiating the surrogate loss $n$ times should give estimates of the $n$th-order derivative, which are used in for example reinforcement learning [15, 14] and meta-learning [12, 27].

5. Variance reduction methods through better sampling and control variates should generalize in higher-order derivative estimation.

6. *Storchastic* should be provably unbiased. To reduce the effort of developing new methods, researchers should only have to prove a set of local conditions that generalize to any SCG.

## 3.2 Gradient Estimators in Storchastic

Next, we introduce each of the four components and motivate why each is needed to ensure Requirement 2 is satisfied. First, we note that several recent gradient estimators, like MAPO [28], unordered set estimator [25] and self-critical baselines [21, 42] act on sequences of stochastic nodes instead of on a single stochastic node. Therefore, we create a partition $\mathcal{S}_1, ..., \mathcal{S}_k$ of $\mathcal{S}_{\prec C}$ topologically ordered by the influence relation, and define the shorthand $\mathbf{x}_i := \mathbf{x}_{\mathcal{S}_i}$. For each partition $\mathcal{S}_i$, we choose a *gradient estimator*, which is a 4-tuple $\langle q_i, w_i, l_i, a_i \rangle$. Here, $q(\mathcal{X}_i|\mathbf{x}_{<i})$ is the *proposal distribution*, $w_i(\mathbf{x}_i)$ is the *weighting function*, $l_i(\mathbf{x}_i)$ is the *gradient function* and $a_i$ is the *control variate*.

### 3.2.1 Proposal distribution

Many gradient estimation methods in the literature do not sample a single value $\mathbf{x}_i \sim p(\mathbf{x}_i|\mathbf{x}_{<i})$, but sample, often multiple, values from possibly a different distribution. Some instances of sampling schemes are taking multiple i.i.d. samples, importance sampling [32] which is very common in off-policy reinforcement learning, sampling without replacement [25], memory-augmented sampling [28] and antithetic sampling [51]. Furthermore, measure-valued derivatives [37, 18] and SPSA [46] also sample from different distributions by comparing the performance of two related distributions. To capture this, the proposal distribution $q(\mathcal{X}_i|\mathbf{x}_{<i})$ samples a *set* of values $\mathcal{X}_i = \{\mathbf{x}_{i,1}, ..., \mathbf{x}_{i,m}\}$ where each $\mathbf{x}_{i,j} \in \Omega_{\mathcal{S}_i}$. The sample is conditioned on $\mathbf{x}_{<i} = \cup_{S \in \mathcal{S}_i} \mathbf{x}_{\mathrm{pa}(S)}$, the values of the parent nodes of the stochastic nodes in $\mathcal{S}_i$. This is illustrated in Figure 1.b.

### 3.2.2 Weighting function

When a gradient estimator uses a different sampling scheme, we have to weight each individual sample to ensure it remains a valid estimate of the expectation. For this, we use a nonnegative weighting function $w_i : \Omega_{\mathcal{S}_i} \to \mathbb{R}^+$. Usually, this function is going to be detached from the computation graph, but we allow it to receive gradients as well to support implementing expectations and gradient estimation methods that compute the expectation over (a subset of) values [25, 28, 30].

### 3.2.3 Gradient function

The gradient function is an unbiased gradient estimator together with the weighting function. It distributes the empirical cost evaluation to the parameters of the distribution. In the case of score function methods, this is the log-probability. For measure-valued derivatives and SPSA we can use the parameters of the distribution itself.

### 3.2.4 Control variate

Modelers can use control variates to reduce the variance of gradient estimates [17, 37]. It is a function that has zero-mean when differentiated. Within the context of score functions, a common control variate is a baseline, which is a function that is independent of the sampled value. We also found that LAX, RELAX, and REBAR (Appendix D.2.4), and the GO gradient [7] (Appendix D.2.6) have natural implementations using a control variate. We discuss how we implement control variates in *Storchastic* in Section 3.5.

### 3.2.5  Example: Leave-one-out baseline

As an example, we show how to formulate the score function with the leave-one-out baseline [35, 22] in *Storchastic*. This method samples $m$ values with replacement and uses the average of the other values as a baseline.

- **Proposal distribution**: We use $m$ samples with replacement, which can be formulated as $q(\mathcal{X}_i|\mathbf{x}_{<i}) = \prod_{j=1}^{m} p(\mathbf{x}_{i,j}|\mathbf{x}_{<i})$.
- **Weighting function**: Since samples are independent, we use $w_i(\mathbf{x}_i) = \frac{1}{m}$.
- **Gradient function**: The score-function uses the log-probability $l_i(\mathbf{x}_i) = \log p(\mathbf{x}_i|\mathbf{x}_{<i})$.
- **Control variate**: We use $a_{i,j}(\mathbf{x}_{<i,j}, \mathcal{X}_i) = (1 - \boxdot(l_i(\mathbf{x}_i)))\frac{1}{m-1}\sum_{j'\neq j} f_C(\mathbf{x}_{<i}, \mathbf{x}_{i,j})$, where $\frac{1}{m-1}\sum_{j'\neq j} f_C(\mathbf{x}_{<i}, \mathbf{x}_{i,j})$ is the leave-one-out baseline. $(1 - \boxdot(l_i(\mathbf{x}_i)))$ is used to ensure the baseline will be subtracted from the cost before multiplication with the gradient function. It will not affect the forward evaluation since $\overrightarrow{1 - \boxdot(l_i(\mathbf{x}_i))}$ evaluates to 0.

### 3.3  The *Storchastic* Surrogate Loss

As mentioned in Requirement 3, we would like to define a *surrogate loss*, which we will introduce next. Differentiating this loss $n$ times, and then evaluating the result using an AD library, will give unbiased estimates of the $n$-th order derivative of the parameter $\theta$ with respect to the cost $C$. Furthermore, according to Requirement 1, we assume the modeler has chosen a gradient estimator $\langle q_i, w_i, l_i, a_i \rangle$ for each partition $\mathcal{S}_i$, which can all be different. Then the Storchastic surrogate loss is

$$SL_{\text{Storch}} = \sum_{\mathbf{x}_1 \in \mathcal{X}_1} w_1(\mathbf{x}_1)\Big[a_1(\mathbf{x}_1, \mathcal{X}_i) + \sum_{\mathbf{x}_2 \in \mathcal{X}_2} w_2(\mathbf{x}_2)\Big[\boxdot(l_1(\mathbf{x}_1))a_2(\mathbf{x}_{<2}, \mathcal{X}_i) + \dots$$
$$+ \sum_{\mathbf{x}_k \in \mathcal{X}_k} w_k(\mathbf{x}_k)\Big[\boxdot(\sum_{j=1}^{k-1} l_j(\mathbf{x}_j))a_k(\mathbf{x}_{<k}, \mathcal{X}_i) + \boxdot(\sum_{i=1}^{k} l_i(\mathbf{x}_i))C\Big]\dots\Big]\Big], \qquad (2)$$
$$\text{where } \mathcal{X}_1 \sim q(\mathcal{X}_1), \mathcal{X}_2 \sim q(\mathcal{X}_2|\mathbf{x}_1), ..., \mathcal{X}_k \sim q(\mathcal{X}_k|\mathbf{x}_{<k}).$$

When this loss is differentiated $n$ times using AD libraries, it will produce unbiased estimates of the $n$-th derivative, as we will show later. To help understand the *Storchastic* surrogate loss and why

---

**Algorithm 1** The *Storchastic* framework: Compute a Monte Carlo estimate of the $n$-th order gradient given $k$ gradient estimators $\langle q_i, w_i, l_i, a_i \rangle$.

---

1: **function** ESTIMATE_GRADIENT($n, \theta$)
2: $\quad SL_{\text{Storch}} \leftarrow \text{SURROGATE\_LOSS}(1, \{\}, 0)$  ▷ Compute surrogate loss
3: $\quad$ **return** $\overrightarrow{\nabla_\theta^{(n)}} SL_{\text{Storch}}$  ▷ Differentiate and use AD library to evaluate surrogate loss
4:
5: **function** SURROGATE_LOSS($i, \mathbf{x}_{<i}, L$)
6: $\quad$ **if** $i = k + 1$ **then**
7: $\quad\quad$ **return** $\boxdot(L)f_C(\mathbf{x}_{\leq k})$  ▷ Use MagicBox to distribute cost
8: $\quad \mathcal{X}_i \sim q(\mathcal{X}_i|\mathbf{x}_{<i})$  ▷ Sample from proposal distribution
9: $\quad \text{sum} \leftarrow 0$
10: $\quad$ **for** $\mathbf{x}_i \in \mathcal{X}_i$ **do**  ▷ Iterate over options in sampled set
11: $\quad\quad A \leftarrow \boxdot(L)a_i(\mathbf{x}_{<i}, \mathcal{X}_i)$  ▷ Compute control variate
12: $\quad\quad L_i \leftarrow L + l_i(\mathbf{x}_i)$  ▷ Compute gradient function
13: $\quad\quad \hat{C} \leftarrow \text{SURROGATE\_LOSS}(i + 1, \mathbf{x}_{\leq i}, L_i)$  ▷ Compute surrogate loss for $\mathbf{x}_i$
14: $\quad\quad \text{sum} \leftarrow \text{sum} + w_i(\mathbf{x}_i)(\hat{C} + A)$  ▷ Weight and add
15: $\quad$ **return** sum

---

it satisfies the requirements, we will break it down using Algorithm 1. The ESTIMATE_GRADIENT function computes the surrogate loss for the SCG, and then differentiates it $n \geq 0$ times using the AD library to return an estimate of the $n$-th order gradient, which should be unbiased according to Requirement 4. If $n$ is set to zero, this returns an estimate of the expected cost.

The SURROGATE_LOSS function computes the equation using a recursive computation, which is illustrated in Figure 1.b-d. It iterates through the partitions and uses the gradient estimator to sample and compute the output of each component. It receives three inputs: The first input $i$ indexes the partitions and gradient estimators, the second input $\mathbf{x}_{<i}$ is the set of previously sampled values for partitions $\mathcal{S}_{<i}$, and $L$ is the sum of gradient functions of those previously sampled values. In line 8, we sample a set of values $\mathcal{X}_i$ for partition $i$ from $q(\mathcal{X}_i|\mathbf{x}_{<i})$. In lines 9 to 14, we compute the sum over values $\mathbf{x}_i$ in $\mathcal{X}_i$, which reflects the $i$-th sum of the equation. Within this summation, in lines 11 and 12, we compute the gradient function and control variate for each value $\mathbf{x}_i$. We will explain in Section 3.5 why we multiply the control variate with the MagicBox of the sum of the previous gradient function.

In line 13, we go into recursion by moving to the next partition. We condition the surrogate loss on the previous samples $\mathbf{x}_{<i}$ together with the newly sampled value $\mathbf{x}_i$. We pass the sum of gradient functions for later usage in the recursion. Finally, in line 14, the sample performance and the control variate are added in a weighted sum. The recursion call happens for each $\mathbf{x}_i \in \mathcal{X}_i$, meaning that this computation is exponential in the size of the sampled sets of values $\mathcal{X}_i$. For example, the surrogate loss samples $|\mathcal{X}_1|$ times from $q_2$, one for each value $\mathbf{x}_1 \in \mathcal{X}_1$. However, this computation can be trivially parallelized by using tensor operations in AD libraries. An illustration of this parallelized computation is given in Figure 1.

Finally, in line 7 after having sampled values for all $k$ partitions, we compute the cost, and multiply it with the MagicBox of the sum of gradient functions. This is similar to what happens in the DiCE estimator in Equation (1). *Storchastic* can be extended to multiple cost nodes by computing surrogate losses for each cost node, and adding these together before differentiation. For stochastic nodes that influence multiple cost nodes, the algorithm can share samples and gradient estimation methods to reduce overhead.

### 3.4 Conditions for Unbiased Estimation

We next introduce our main result that shows *Storchastic* satisfies Requirements 4 and 6, namely the conditions the gradient estimators should satisfy such that the *Storchastic* surrogate loss gives estimates of the $n$-th order gradient of the SCG. A useful part of our result is that, in line with Requirement 6, only local conditions of gradient estimators have to be proven to ensure estimates are unbiased. Our result gives immediate generalization of these local proofs to any SCG.

**Theorem 1.** *Evaluating the $n$-th order derivative of the* Storchastic *surrogate loss in Equation* (2) *using an AD library is an unbiased estimate of* $\nabla_\theta^{(n)}\mathbb{E}[C]$ *under the following conditions. First, all functions $f_F$ corresponding to deterministic nodes $F$ and all probability measures $p_S$ corresponding to stochastic nodes $S$ are* identical under evaluation. *Secondly, for each gradient estimator* $\langle q_i, w_i, l_i, a_i \rangle$, $i = 1, ..., k$, *all the following hold for $m = 0, ..., n$:*

1. $\mathbb{E}_{q(\mathcal{X}_i|\mathbf{x}_{<i})}[\sum_{\mathbf{x}_i \in \mathcal{X}_i} \overrightarrow{\nabla_\theta^{(m)} w_i(\mathbf{x}_i)\boxdot(l_i(\mathbf{x}_i))f(\mathbf{x}_i)}] = \overrightarrow{\nabla_\theta^{(m)}\mathbb{E}_{\mathcal{S}_i}[f(\mathbf{x}_i)]}$ *for any deterministic function $f$;*

2. $\mathbb{E}_{q(\mathcal{X}_i|\mathbf{x}_{<i})}[\sum_{\mathbf{x}_i \in \mathcal{X}_i} \overrightarrow{\nabla_\theta^{(m)} w_i(\mathbf{x}_i)a_i(\mathbf{x}_{<i}, \mathcal{X}_i)}] = 0$;

3. *for $n \geq m > 0$,* $\mathbb{E}_{q(\mathcal{X}_i|\mathbf{x}_{<i})}[\sum_{\mathbf{x}_i \in \mathcal{X}_i} \overrightarrow{\nabla_\theta^{(m)} w_i(\mathbf{x}_i)}] = 0$;

4. $\overrightarrow{q(\mathcal{X}_i|\mathbf{x}_{<i})} = q(\mathcal{X}_i|\mathbf{x}_{<i})$, *for all permissible $\mathcal{X}_i$.*

The first condition defines a local surrogate loss for single expectations of any function under the proposal distribution. The condition then says that this surrogate loss should give an unbiased estimate of the gradient for all orders of differentiation $m = 0, ..., n$. Note that since 0 is included, the forward evaluation should also be unbiased. This is the main condition used to prove unbiasedness of the *Storchastic* framework, and can be proven for the score function and expectation, and for measure-valued derivatives and SPSA for zeroth and first-order differentiation.

The second condition says that the control variate should be 0 in expectation under the proposal distribution for all orders of differentiation. This is how control variates are defined in previous work [37], and should usually not restrict the choice. The third condition constrains the weighting

function to be 0 in expectation for orders of differentiation larger than 0. Usually, this is satisfied by the fact that weighting functions are detached from the computation graph, but when enumerating expectations, this can be shown by using that the sum of weights is constant. The final condition is a regularity condition that says proposal distributions should not be different under forward mode. We also assume that the SCG is *identical under evaluation*. This means that all functions and probability densities evaluate to the same value with and without the forward-mode operator, even when differentiated. This concept is formally introduced in Appendix A.

A full formalization and the proof of Theorem 1 are given in Appendix B.1. The general idea is to rewrite each sampling step as an expectation, and then inductively show that the inner expectation $i$ over the proposal distribution $q_i$ is an unbiased estimate of the $n$th-order derivative over $\mathcal{S}_i$ conditional on the previous samples. To reduce the multiple sums over gradient functions inside MagicBox, we make use of a property of MagicBox proven in Appendix A:

**Proposition 4.** *Summation inside a MagicBox is equivalent under evaluation to multiplication of the arguments in individual MagicBoxes, ie:*

$$\boxdot(l_1(x) + l_2(x))f(x) \overset{\Rrightarrow}{\equiv} \boxdot(l_1(x))\boxdot(l_2(x))f(x).$$

*Equivalence under evaluation*, denoted $\overset{\Rrightarrow}{\equiv}$, informally means that, under evaluation of $\to$, the two expressions and their derivatives are equal. This equivalence is closely related to $e^{a+b} = e^a e^b$.

## 3.5 Any-order variance reduction using control variates

To satisfy Requirement 5, we investigate implementing control variates such that the variance of any-order derivatives is properly decreased. This is challenging in general SCG's [33], since in higher orders of differentiation, derivatives of gradient functions will interact, but naive implementations of control variates only reduce the variance of the gradient function corresponding to a single stochastic node. *Storchastic* solves this problem similarly to the method introduced in [33]. In line 11 of the algorithm, we multiply the control variate with the sum of preceding gradient functions $\boxdot(L)$. We prove that this ensures every term of the any-order derivative will be affected by a control variate in Appendix C. This proof is new, since [33] only showed this for first and second order differentiation, not for general control variates, and uses a slightly different formulation that we show misses some terms.

**Theorem 2** (informal). *Let $L_i = \sum_{j=1}^{i} l_i$. The* Storchastic *surrogate loss of* (2) *can equivalently be computed as*

$$SL_{\text{Storch}} \overset{\Rrightarrow}{\equiv} \sum_{\mathbf{x}_1 \in \mathcal{X}_1} \sum_{\mathbf{x}_2 \in \mathcal{X}_2} \cdots \sum_{\mathbf{x}_k \in \mathcal{X}_k} \prod_{i=1}^{k} w_i(\mathbf{x}_i) \sum_{i=1}^{k} \boxdot(L_{i-1}) \Big( a_i(\mathbf{x}_{<i}, \mathcal{X}_i) + (\boxdot(l_i) - 1)C \Big) + C.$$

This gives insight into how control variates are used in *Storchastic*. They are added to the gradient function, but only during differentiation since $\overrightarrow{\boxdot(L_i) - 1} = 0$. Furthermore, since both terms are multiplied with $\boxdot(L_{i-1})$ (see line 11 of Algorithm 1), both terms correctly distribute over the same any-order derivative terms. By choosing a control variate of the form $a_i(\mathbf{x}_{<i}, \mathcal{X}_i) = (1 - \boxdot(l_i)) \cdot b_i$, we recover baselines which are common in the context of score functions [13, 37]. For the proof, we use the following proposition also proven in Appendix C:

**Proposition 7.** *For orders of differentiation $n > 0$,*

$$\overrightarrow{\nabla_N^{(n)}\boxdot(L_k)} = \nabla_N^{(n)} \overrightarrow{\sum_{i=1}^{k} (\boxdot(l_i) - 1)\boxdot(L_{i-1})}.$$

## 3.6 Gradient Estimation Methods

In Appendix D we show how several prominent examples of gradient estimation methods in the literature can be formulated using *Storchastic*, and prove for what orders of differentiation the conditions hold. Starting off, we show that for finite discrete random variables, we can formulate enumerating over all possible options using *Storchastic*. The score function fits by mimicking DiCE [13]. We extend it to multiple samples with replacement to allow using the leave-one-out baseline [35, 22].

```
1  class ScoreFunctionLOO(storch.method.Method):
2      def proposal_dist(self, distribution, amt_samples):
3          return distr.sample((amt_samples,))
4
5      def weighting_function(self, distribution, amt_samples):
6          return torch.full(amt_samples, 1/amt_samples)
7
8      def estimator(self, sample, cost):
9          # Compute gradient function (log-probability)
10         log_prob = sample.distribution.log_prob(tensor)
11         sum_costs = storch.sum(costs.detach(), sample.name)
12         # Compute control variate
13         baseline = (sum_costs - costs) / (sample.n - 1)
14         return log_prob, (1.0 - magic_box(log_prob)) * baseline
```

Figure 3: Implementing the score function with the leave-one-out baseline in the Storchastic library.

Furthermore, we show how importance sampling, sum-and-sample estimators such as MAPO [28], the unordered set estimator [25] and RELAX and REBAR [16, 47] fit in *Storchastic*. We also discuss the antithetic sampling estimator ARM [51]. Unfortunately, condition 2 only holds for this estimator for $n \leq 1$ since it relies on a particular property of the score function that holds only for first-order gradient estimation. In addition to score function based methods, we discuss the GO gradient, SPSA [44] and Measure-Valued Derivative [18], and show that the last two will only be unbiased for $n \leq 1$. Finally, we note that reparameterization [20, 43] can be implemented by transforming the SCG such that the sampling step is outside the path from the parameter to the cost [45].

## 3.7 Implementation

We implemented *Storchastic* as an open source PyTorch [39] library [1]. To ensure modelers can easily use this library, it automatically handles sets of samples as extra dimensions to PyTorch tensors which allows running multiple sample evaluations in parallel. This approach is illustrated in Figure 1. By making use of PyTorch broadcasting semantics, this allows defining models for simple single-sample computations that are automatically parallelized using *Storchastic* when using multiple samples. The *Storchastic* library has implemented most of the gradient estimation methods mentioned in Section 3.6. Furthermore, new gradient estimation methods can seamlessly be added.

### 3.7.1 Example: Leave-one-out baseline in Discrete Variational Autoencoder

As a small case study, we show how to implement the score function with the leave-one-out baseline introduced in Section 3.2.5 for the discrete variational autoencoder introduced in Section 2.3 in PyTorch using Storchastic. While the code listed is simplified, it shows the flexibility with which one can compute gradients in SCGs.

We list in Figure 3 how to implement the score function with the leave-one-out baseline. Line 3 implements the proposal distribution, line 6 the weighting function, line 10 the gradient function and line 13 and 14 the control variate. Gradient estimation methods in Storchastic all extend a common base class `storch.method.Method` to allow easy interoperability between different methods.

In Figure 4, we show how to implement the discrete VAE. The implementation directly follows the SCG shown in Figure 2. In line 2, we create the `ScoreFunctionLOO` method defined in Figure 3. Then, we run the training loop: In line 6 we create the stochastic node $x$ by denoting the minibatch dimension as an independent dimension. In line 8 we run the encoder with parameters $\phi$ to find the variational posterior $q_z$. We call the gradient estimation method in line 9 to get a sample of $z$. Note that this interface is independent of gradient estimation method chosen, meaning that if we wanted to compare our implemented method with a baseline, all that is needed is to change line 2. After the decoder, we compute the two costs in lines 12 and 13. Finally, we call Storchastic main algorithm in line 15 and run the optimizer.

---

[1]Code is available at `github.com/HEmile/storchastic`.

```
1  from vae import minibatches, encode, decode, KLD, binary_cross_entropy
2  method = ScoreFunctionLOO("z", 8)
3  for data in minibatches():
4      optimizer.zero_grad()
5      # Denote minibatch dimension as independent plate dimension
6      data = storch.denote_independent(data.view(-1, 784), 0, "data")
7      # Compute variational distribution given data, sample z
8      q = torch.distributions.OneHotCategorical(logits=encode(data))
9      z = method(q)
10     # Compute costs, form the ELBO
11     reconstruction = decode(z)
12     storch.add_cost(KLD(q))
13     storch.add_cost(binary_cross_entropy(reconstruction, data))
14     # Storchastic backward pass, optimize
15     ELBO = storch.backward()
16     optimizer.step()
```

Figure 4: Simplified implementation of the discrete VAE using Storchastic.

We run this model on our currently implemented set of gradient estimation methods for discrete variables in Appendix E and report the results, which are meant purely to illustrate the case study.

## 4  Related Work

The literature on gradient estimation is rich, with papers focusing on general methods that can be implemented in *Storchastic* [46, 18, 16, 51, 28, 30, 7], see Appendix D, and works focused on Reinforcement Learning [49, 29, 36] or Variational Inference [35]. For a recent overview, see [37].

The literature focused on SCGs is split into methods using reparameterization [43, 20, 11, 31, 19] and those using the score function [45]. Of those, DiCE [13] is most similar to *Storchastic*, and can do any-order estimation on general SCGs. DiCE is used in the probabilistic programming library Pyro [3]. We extend DiCE to allow for incorporating many other gradient estimation methods than just basic score function. We also derive and prove correctness of a general implementation for control variates for any-order estimation which is similar to the one conjectured for DiCE in [33].

[38, 50] and [48] study actor-critic-like techniques and bootstrapping for SCGs to incorporate reparameterization using methods inspired by deterministic policy gradients [29]. By using models to differentiate through, these methods are biased through model inaccuracies and thus do not directly fit into *Storchastic*. However, combining these ideas with the automatic nature of *Storchastic* could be interesting future work.

## 5  Conclusion

We investigated general automatic differentiation for stochastic computation graphs. We developed the *Storchastic* framework, and introduced an algorithm for unbiased any-order gradient estimation that allows using a large variety of gradient estimation methods from the literature. We also investigated variance reduction and showed how to properly implement control variates such that it affects any-order gradient estimates. The framework satisfies the requirements introduced in Section 3.1.

For future work, we are interested in extending the analysis of *Storchastic* to how variance compounds when using different gradient estimation methods. Furthermore, *Storchastic* could be extended to allow for biased methods. We are also interested in closely analyzing the different components of gradient estimators, both from a theoretical and empirical point of view, to develop new estimators that combine the strengths of estimators in the literature.

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
