## A Forward-mode evaluation

In this section, we define several operators that we will use to mathematically define operators used within deep learning to implement gradient estimators.

To define these, we will need to distinguish how deep learning libraries evaluate their functions. [13] handles this using a different kind of equality, denoted $\mapsto$. Unfortunately, it is not formally introduced, making it unclear as to what rules are allowed with this equality. For instance, they define the DiCE operator as

$$1.\ \boxdot(f(\theta)) \mapsto 1$$
$$2.\ \nabla_\theta \boxdot(f(\theta)) = \boxdot(f(\theta))\nabla_\theta f(\theta)$$

However, without a clearly defined meaning of $\mapsto$ 'equality under evaluation', it is unclear whether the following is allowed:

$$\nabla_\theta \boxdot(f(\theta)) \mapsto \nabla_\theta 1 = 0$$

This would lead to a contradiction, as by definition

$$\nabla_\theta \boxdot(f(\theta)) = \boxdot(f(\theta))\nabla_\theta f(\theta) \mapsto \nabla_\theta f(\theta)$$

. We first introduce an unambigiuous formulation for forward mode evaluation that does not allow such inconsistencies.

**Definition 2.** The *stop-grad* operator $\perp$ is a function such that $\nabla_x \perp(x) = 0$. The *forward-mode* operator $\rightarrow$ is a function such that, for well formed formulas $a$ and $b$,

1. $\overrightarrow{\perp(a)} = \vec{a}$

2. $\overrightarrow{a+b} = \vec{a} + \vec{b}$

3. $\overrightarrow{a \cdot b} = \vec{a} \cdot \vec{b}$

4. $\overrightarrow{a^b} = \vec{a}^{\vec{b}}$

5. $\vec{c} = c$, if $c$ is a constant or a variable.

6. $\overrightarrow{\vec{a}} = \vec{a}$

Additionally, we define the DiCE operator $\boxdot(x) = \exp(x - \perp(x))$

When computing the results of a function $f(x)$, Deep Learning libraries instead compute $\overrightarrow{f(x)}$. Importantly, $\overrightarrow{\nabla_x f(x)}$ does not always equal $\nabla_x \overrightarrow{f(x)}$. For example, $\overrightarrow{\nabla_x \perp(f(x))} = \vec{0} = 0$, while $\nabla_x \overrightarrow{\perp(f(x))} = \nabla_x \overrightarrow{f(x)}$.

In the last example, the derivative will first have to be rewritten to find a closed-form formula that does not contain the $\rightarrow$ operator. Furthermore, $\perp(f(x))$ only evaluates to a closed-form formula if it is reduced using derivation, or if it is enclosed in $\rightarrow$.

We note that $\mathbb{E}_{p(x)}[\overrightarrow{f(x)}] = \overrightarrow{\mathbb{E}_{p(x)}[f(x)]}$ for both continuous and discrete distributions $p(x)$ if $\overrightarrow{p(x)} = p(x)$. This is easy to see for discrete distributions since these are weighted sums over an amount of elements. For continuous distributions we can use the Riemann integral definition.

$$\mathbb{E}_{p(x)}[\overrightarrow{f(x)}] = \int p(x)\overrightarrow{f(x)}$$

**Proposition 1.** *1:* $\overrightarrow{\boxdot(f(x))} = 1$ *and 2:* $\nabla_x \boxdot(f(x)) = \boxdot(f(x)) \cdot \nabla_x f(x)$

*Proof.* 1.

$$\overrightarrow{\boxdot(f(x))} = \overrightarrow{\exp(f(x) - \perp(f(x)))}$$
$$= \exp(\overrightarrow{f(x)} - \overrightarrow{\perp(f(x))})$$
$$= \exp(f(x) - f(x)) = 1$$

2.

$$\nabla_x \boxdot(f(x)) = \nabla_x \exp(f(x) - \bot(f(x)))$$
$$= \exp(f(x) - \bot(f(x)))\nabla_x(f(x) - \bot(f(x)))$$
$$= \boxdot(f(x))(\nabla_x f(x) - \nabla_x \bot(f(x))\nabla_x f(x))$$
$$= \boxdot(f(x))(\nabla_x f(x) - 0 \cdot \nabla_x f(x)) = \boxdot(f(x))\nabla_x f(x)$$

$\square$

Furthermore, unlike in the DiCE paper, with this notation $\overrightarrow{\nabla_x \boxdot(f(x))}$ unambiguously evaluates to $\overrightarrow{\nabla_x f(x)}$, as $\overrightarrow{\nabla_x \boxdot(f(x))} = \overrightarrow{\boxdot(f(x))\nabla_x f(x)} = \overrightarrow{\boxdot(f(x))} \cdot \overrightarrow{\nabla_x f(x)} = \overrightarrow{\nabla_x f(x)}$. Note that, although this is not a closed-form formula, by finding a closed-form formula for $\nabla_x f(x)$, this can be reduced to $\nabla_x f(x)$.

**Proposition 2.** *For any two functions $f(x)$ and $l(x)$, it holds that for all $n \in (0, 1, 2, ...)$,*

$$\overrightarrow{\nabla_x^{(n)} \boxdot(l(x)f(x))} = \overrightarrow{g^{(n)}(x)}.$$

*where $g^{(n)}(x) = \nabla_x g^{(n-1)}(x) + g^{(n-1)}(x)\nabla_x l(x)$ for $n > 0$, and $g^{(0)}(x) = f(x)$.*

For this proof, we use a similar argument as in [13].

*Proof.* First, we show that $\boxdot(l(x))g^{(n)}(x) = \nabla_x^{(n)}\boxdot(l(x))f(x)$. We start off with the base case, $n = 0$. Then, $\boxdot(l(x))g^{(0)}(x) = \boxdot(l(x))f(x)$.

Next, assume the proposition holds for $n$, that is, $\boxdot(l(x))g^{(n)}(x) = \nabla_x^{(n)}\boxdot(l_i)f(x)$. Consider $n+1$.

$$\boxdot(l(x))g^{(n+1)}(x) = \boxdot(l(x))(\nabla_x g^{(n)}(x) + g^{(n)}(x)\nabla_x l(x))$$
$$= \nabla_x \boxdot(l(x))g^{(n)}(x)$$
$$= \nabla_x(\nabla_x^{(n)}(\boxdot(l(x))f(x)))$$
$$= \nabla_x^{(n+1)}\boxdot(l(x))f(x)$$

Where from line 1 to 2 we use the DiCE proposition in the reversed direction. From 2 to 3 we use the inductive hypothesis.

We use this result, $\boxdot(l(x))g^{(n)}(x) = \nabla_x^{(n)}\boxdot(l(x))f(x)$, to prove our proposition. Since $\overrightarrow{a} = 1 \cdot \overrightarrow{a} = \overrightarrow{\boxdot(l_i)}\overrightarrow{a} = \overrightarrow{\boxdot(l_i)a}$,

$$\overrightarrow{g^{(n)}(x)} = \overrightarrow{\boxdot(l(x))g^{(n)}(x)} = \overrightarrow{\nabla_N^{(n)}\boxdot(l(x))f(x)}$$

$\square$

**Definition 3.** We say a function $f$ is *identical under evaluation* if for all $n \in (0, 1, 2, ...)$, $\overrightarrow{\nabla_x^{(n)} f(x)} = \nabla_x^{(n)} f(x)$. Furthermore, we say that two functions $f$ and $g$ are *equivalent under evaluation*, denoted $f \stackrel{\Rrightarrow}{=} g$, if for all $n \in (0, 1, 2, ...)$, $\overrightarrow{\nabla_x^{(n)} f(x)} = \overrightarrow{\nabla_x^{(n)} g(x)}$.

Every function that does not contain a stop-grad operator ($\bot$) is identical under evaluation, although functions that are identical under evaluation can have stop-grad operators (for example, consider $f(x) \stackrel{\Rrightarrow}{=} f(x) + \bot(f(x) - f(x))$). Note that $\overrightarrow{f(x)} = f(x)$ does not necessarily mean that $f$ is identical under evaluation, since for instance the function $f'(x) = \boxdot(2x)f(x)$ has $\overrightarrow{f'(x)} = f(x)$, but $\overrightarrow{\nabla_x f'(x)} = \overrightarrow{\boxdot(2x)(\nabla_x f(x) + 2)} = \nabla_x f(x) + 2 \neq \nabla_x f'(x) = \boxdot(2x)(\nabla_x f(x) + 2)$.

**Proposition 3.** *If $f(x)$ and $l(x)$ are identical under evaluation, then all $g^{(n)}(x)$ from $n = 0, ..., n$ as defined in Proposition 2 are also identical under evaluation.*

*Proof.* Consider $n = 0$. Then $g^{(0)}(x) = f(x)$. Since $f(x)$ is identical under evaluation, $g^{(0)}$ is as well.

Assume the proposition holds for $n$, and consider $n + 1$. Let $m$ be any positive number. $\overrightarrow{\nabla_x^{(m)} g^{(n+1)}(x)} = \overrightarrow{\nabla_x^{(m)}(\nabla_x g^{(n)}(x) + g^{(n)}(x)\nabla_x l(x))}$. Since $g^{(n)}(x)$ is identical under evaluation by the inductive hypothesis, $\overrightarrow{\nabla_x^{(m)}\nabla_x g^{(n)}(x)} = \overrightarrow{\nabla_x^{(m+1)} g^{(n)}(x)} = \nabla_x^{(m+1)} g^{(n)}(x)$.

Next, using the general Leibniz rule, we find that $\overrightarrow{\nabla_x^{(m)} g^{(n)}(x)\nabla_x l(x)} = \sum_{j=0}^m \binom{m}{j \overrightarrow{\nabla_x^{(m-j)} g^{(n)}(x)}}\overrightarrow{\nabla_x^{(j+1)} l(x)}$. Since both $g^{(n)}(x)$ and $l(x)$ are identical under evaluation, this is equal to $\sum_{j=0}^m \binom{m}{j}\nabla_x^{(m-j)} g^{(n)}(x)\nabla_x^{(j+1)} l(x)) = \nabla_x^{(m)} g^{(n)}(x)\nabla_x l(x)$.

Therefore, $\overrightarrow{\nabla_x^{(m)} g^{(n+1)}(x)} = \nabla_x^{(m)}(\nabla_x g^{(n)}(x) + g^{(n)}(x)\nabla_x l(x)) = \nabla_x^{(m)} g^{(n+1)}(x)$, which shows that $g^{(n+1)}(x)$ is identical under evaluation. $\qquad\square$

We next introduce a very useful proposition that we will use to prove unbiasedness of the *Storchastic* framework. This result was first used without proof in [10].

**Proposition 4.** *For any three functions* $l_1(x)$, $l_2(x)$ *and* $f(x)$, $\boxdot(l_1(x) + l_2(x))f(x) \overset{\rightrightarrows}{\equiv} \boxdot(l_1(x))\boxdot(l_2(x))f(x)$. *That is, for all* $n \in (0, 1, 2, ...)$.

$$\overrightarrow{\nabla_x^{(n)}\boxdot(l_1(x) + l_2(x))f(x)} = \overrightarrow{\nabla_x^{(n)}\boxdot(l_1(x))\boxdot(l_2(x))f(x)}$$

*Proof.* Start with the base case $n = 0$. Then, $\overrightarrow{\boxdot(l_1(x) + l_2(x))f(x)} = \overrightarrow{f(x)} = 1 \cdot 1 \cdot \overrightarrow{f(x)} = \overrightarrow{\boxdot(l_1(x))\boxdot(l_2(x))f(x)}$.

Next, assume the proposition holds for $n$. Then consider $n + 1$:

$$\overrightarrow{\nabla_x^{(n+1)}\boxdot(l_1(x))\boxdot(l_2(x))f(x)}$$
$$= \overrightarrow{\nabla_x^{(n)}\nabla_x\boxdot(l_1(x)\boxdot(l_2(x))f(x))}$$
$$= \overrightarrow{\nabla_x^{(n)}\boxdot(l_1(x))\boxdot(l_2(x))(f(x)\nabla_x l_1(x) + f(x)\nabla_x l_2(x) + \nabla_x f(x))}$$
$$= \overrightarrow{\nabla_x^{(n)}\boxdot(l_1(x))\boxdot(l_2(x))(f(x)\nabla_x(l_1(x) + l_2(x)) + \nabla_x f(x))}$$

Define function $h(x) = f(x)\nabla_x(l_1(x) + l_2(x)) + \nabla_x f(x)$. Since the proposition works for any function, we can apply the inductive hypothesis replacing $f(x)$ by $h(x)$:

$$\overrightarrow{\nabla_x^{(n)}\boxdot(l_1(x))\boxdot(l_2(x))h(x)}\overrightarrow{\nabla_x^{(n)}\boxdot(l_1(x) + l_2(x))h(x)}$$

Finally, we use Proposition 2 with $g^{(1)}(x) = h(x)$ and $l(x) = l_1(x) + l_2(x)$:

$$\overrightarrow{\nabla_x^{(n)}\boxdot(l_1(x) + l_2(x))f(x)\nabla_x(l_1(x) + l_2(x)) + \nabla_x f(x)}$$
$$= \overrightarrow{\nabla_x^{(n)}\nabla_x\boxdot(l_1(x) + l_2(x))f(x)} = \overrightarrow{\nabla_x^{(n+1)}\boxdot(l_1(x) + l_2(x))f(x)}$$

$$\square$$

It should be noted that it cannot be proven that $\nabla_x^{(n)}\boxdot(\sum_{i=1}^k l_i(x))f(x) = \nabla_x^{(n)}\boxdot(\sum_{i=1}^{k-1} l_i(x))\boxdot(l_k(x))f(x)$ because the base-case cannot be proven without the $\rightarrow$ operator interpreting the $\boxdot$ operator.

Also note the parallels with the exponential function, where $e^{l_1(x)+l_2(x)} = e^{l_1(x)}e^{l_2(x)}$.

# B The *Storchastic* framework (formal)

In this section we formally introduce *Storchastic* to provide the mathematical machinery needed to prove our results. Let $\mathcal{S}_1, \ldots, \mathcal{S}_k$ be a partition of $\mathcal{S}_{\prec F}$. Assume the sets $\mathcal{S}_1, \ldots, \mathcal{S}_k$ are topologically sorted, that is, there is no $i < j$ such that there exists a stochastic node $S \in \mathcal{S}_j$ that is also in $\mathcal{S}_{<i} = \bigcup_{j=1}^{j-1} \mathcal{S}_j$. We use assignment $\mathbf{x}_i$ to denote a set that gives a value to each of the random variables $S \in \mathcal{S}_i$. That is, $\mathbf{x}_i \in \prod_{S \in \mathcal{S}_i} \Omega_S$. We additionally use $\mathbf{x}_{<i}$ to denote a set that gives values to all random variables in $\mathcal{S}_{<i}$. In the same vein, $\mathcal{X}_i$ denotes a set of sets of values $\mathbf{x}_i$, that is $\mathcal{X}_i = \{\mathbf{x}_{i,1}, ..., \mathbf{x}_{i,|\mathcal{X}_i|}\}$.

**Definition 4.** For each partition $\mathcal{S}_i$ there is a **gradient estimator** $\langle q_i, w_i, l_i, a_i \rangle$ where $q(\mathcal{X}_i|\mathbf{x}_{<i})$ is a distribution over a set of values $\mathcal{X}_i$ conditioned on $\mathbf{x}_{<i}$, $w_i : \prod_{S \in \mathcal{S}_i} \Omega_S \to \mathbb{R}^+$ is the weighting function that weights different values $\mathbf{x}_i$, $l_i : \prod_{S \in \mathcal{S}_i} \Omega_S \to \mathbb{R}$ is the gradient function that provides the gradient produced by each $\mathbf{x}_i$, and the control variate $a_i : \prod_{j=1}^i \prod_{S \in \mathcal{S}_j} \Omega_S \to \mathbb{R}$ is a function of both $\mathbf{x}_i$ and $\mathbf{x}_{<i}$.

$q(\mathcal{X}_i|\mathbf{x}_{<i})$ is factorized as follows: Order stochastic nodes $S_{i,1}, \ldots, S_{i,m} \in \mathcal{S}_i$ topologically, then $q(\mathcal{X}_i|\mathbf{x}_{<i}) = \prod_{j=1}^m q(\mathcal{X}_{i,j}|\mathcal{X}_{i,<j}, \mathbf{x}_{<i})$.

In the rest of this appendix, we will define some shorthands to declutter the notation, as follows:

- $w_i = w_i(\mathbf{x}_i)$ and $W_i = \prod_{j=1}^i w_i$
- $l_i = l_i(\mathbf{x}_i)$ and $L_i = \sum_{j=1}^i l_i$
- $a_i = a_i(\mathbf{x}_{<i}, \mathcal{X}_i)$
- $q_1 = q(\mathcal{X}_1)$ and $q_i = q(\mathcal{X}_i|\mathbf{x}_{<i})$ (for $i > 1$)

These interfere with the functions and distributions themselves, but it should be clear from context which of the two is meant.

**Proposition 5.** *Given a topologically sorted partition $\mathcal{S}_1, ..., \mathcal{S}_k$ of $\mathcal{S}_{\prec F}$ and corresponding gradient estimators $\langle q_i, w_i, l_i, a_i \rangle$ for each $1 \leq i \leq k$, the evaluation of the $n$-th order derivative of the Storchastic surrogate loss $\overrightarrow{\nabla_N^{(n)}} SL_{\mathrm{Storch}}$ of Equation 2 is equal in expectation to*

$$\mathbb{E}_{q_1}\left[ \sum_{\mathbf{x}_1 \in \mathcal{X}_1} \overrightarrow{\nabla_N^{(n)} w_1 a_1} + \ldots \mathbb{E}_{q_k}\left[ \sum_{\mathbf{x}_k \in \mathcal{X}_k} \overrightarrow{\nabla_N^{(n)} W_k \boxdot (L_{k-1}) a_k} + \overrightarrow{\nabla_N^{(n)} W_k \boxdot (L_k) F} \right] \ldots \right] \quad (3)$$

*where the $i$-th term in the dots is $\mathbb{E}_{q_i}[\sum_{\mathbf{x}_i \in \mathcal{X}_i} \overrightarrow{\nabla_N^{(n)} W_i \boxdot L_{i-1} a_i} + (\ldots)]$*

*Proof.* By moving the weights inwards and using the $L_i$ notation,

$$\overrightarrow{\nabla_N^{(n)} SL_{\mathrm{Storch}}} = \overrightarrow{\nabla_N^{(n)} \sum_{\mathbf{x}_1 \in \mathcal{X}_1} w_1 \left[ a_1 + \cdots + \sum_{\mathbf{x}_k \in \mathcal{X}_k} w_k \left[ \boxdot (L_{k-1}) a_k + \boxdot (L_k) C \right] \ldots \right]}$$

$$= \overrightarrow{\nabla_N^{(n)} \sum_{\mathbf{x}_1 \in \mathcal{X}_1} w_1 a_1 + \cdots \sum_{\mathbf{x}_i \in \mathcal{X}_i} W_i \boxdot (L_{i-1}) a_i + \ldots}$$

$$+ \sum_{\mathbf{x}_k \in \mathcal{X}_k} \overrightarrow{W_k \boxdot (L_{k-1}) a_k + W_k \boxdot (L_k) C}$$

$$= \sum_{\mathbf{x}_1 \in \mathcal{X}_1} \overrightarrow{\nabla_N^{(n)} w_1 a_1} + \cdots \sum_{\mathbf{x}_i \in \mathcal{X}_i} \overrightarrow{\nabla_N^{(n)} W_i \boxdot (L_{i-1}) a_i} + \ldots$$

$$+ \sum_{\mathbf{x}_k \in \mathcal{X}_k} \overrightarrow{\nabla_N^{(n)} W_k \boxdot (L_{k-1}) a_k} + \overrightarrow{\nabla_N^{(n)} W_k \boxdot (L_k) C}$$

This is all under sampling $\mathcal{X}_1 \sim q(\mathcal{X}_1), \mathcal{X}_2 \sim q(\mathcal{X}_2|\mathbf{x}_1), ..., \mathcal{X}_k \sim q(\mathcal{X}_k|\mathbf{x}_{<k})$. Taking expectations over these distributions before the respective summation over $\mathcal{X}i$ gives the result. $\qquad\square$

In the *Storchastic* framework, we require that $\mathbb{E}[F]$ is identical under evaluation, that is, $\overrightarrow{\nabla_N^{(n)}\mathbb{E}[F]} = \nabla_N^{(n)}\mathbb{E}[F]$. This in practice means that the probability distributions and functions in the stochastic computation graph contain no stop gradient operators ($\perp$).

Using Proposition 2, we give a recursive expression for $\overrightarrow{\nabla_N^{(n)}w_i\big(a_i + \boxdot(l_i)f(\mathbf{x}_i)\big)}$.

**Proposition 6.** *For any gradient estimator* $\langle q_i, w_i, l_i, a_i \rangle$ *it holds that*

$$\overrightarrow{\nabla_N^{(n)}w_i\big(\boxdot(L_{i-1})a_i + \boxdot(l_i)f(\mathbf{x}_i)\big)} = \overrightarrow{\nabla_N^{(n)}w_i\boxdot(L_{i-1})a_i + g_i^{(n)}(\mathbf{x}_i)}$$

*where* $g_i^{(n)}(\mathbf{x}_i) = \nabla_N g_i^{(n-1)}(\mathbf{x}_i) + g_i^{(n-1)}\nabla_N l_i$ *for* $n > 0$, *and* $g_i^{(0)}(\mathbf{x}_i) = w_i f(\mathbf{x}_i)$.

*Proof.* Using Proposition 2, we find that

$$\overrightarrow{\nabla_N^{(n)}w_i\boxdot(L_{i-1})a_i + g_i^{(n)}(\mathbf{x}_i)} = \overrightarrow{\nabla_N^{(n)}w_i\boxdot(L_{i-1})a_i} + \overrightarrow{\nabla_N^{(n)}w_i\boxdot(l_i)f(\mathbf{x}_i)}$$
$$= \overrightarrow{\nabla_N^{(n)}w_i\big(\boxdot(L_{i-1})a_i + \boxdot(l_i)f(\mathbf{x}_i)\big)}$$

$\square$

Proposition 6 is useful because it gives a fairly simple recursion to proof unbiasedness of any-order estimators with, when the gradient estimator is implemented in *Storchastic*. Note that it doesn't itself show that such gradient estimators are unbiased in any-order derivatives.

## B.1 Unbiasedness of the *Storchastic* framework

In this section, we use the equivalent expectation from Proposition 5

**Theorem 1.** *Let* $\langle q_i, w_i, l_i, a_i \rangle$ *for* $i = 1, ..., k$ *be a sequence of gradient estimators. Let the stochastic computation graph* $\mathbb{E}[F]$ *be identical under evaluation[2]. The evaluation of the* $n$*th-order derivative of the* Storchastic *surrogate loss is an unbiased estimate of* $\nabla_N^{(n)}\mathbb{E}[F]$ *, that is*

$$\nabla_N^{(n)}\mathbb{E}[F] = \mathbb{E}_{q_1}\left[\sum_{\mathbf{x}_1 \in \mathcal{X}_1} \overrightarrow{\nabla_N^{(n)}w_1 a_1} + \ldots \mathbb{E}_{q_k}\left[\sum_{\mathbf{x}_k \in \mathcal{X}_k} \overrightarrow{\nabla_N^{(n)}W_k\boxdot(L_{k-1})a_k} + \overrightarrow{\nabla_N^{(n)}W_k\boxdot(L_k)F}\right]\ldots\right]$$

*if the following conditions hold for all estimators* $i = 1, ..., k$ *and all preceding orders of differentiation* $n \geq m \geq 0$*:*

1. $\mathbb{E}_{q_i}[\sum_{\mathbf{x}_i \in \mathcal{X}_i} \overrightarrow{\nabla_N^{(m)}w_i\boxdot(l_i)f(\mathbf{x}_i)}] = \overrightarrow{\nabla_N^{(m)}\mathbb{E}_{\mathcal{S}_i}[f(\mathbf{x}_i)]}$ *for any deterministic function* $f$*;*

2. $\mathbb{E}_{q_i}[\sum_{\mathbf{x}_i \in \mathcal{X}_i} \overrightarrow{\nabla_N^{(m)}w_i a_i}] = 0$*;*

3. *for* $n \geq m > 0$, $\mathbb{E}_{q_i}[\sum_{\mathbf{x}_i \in \mathcal{X}_i} \overrightarrow{\nabla_N^{(m)}w_i}] = 0$*;*

4. $\overrightarrow{q(\mathcal{X}_i|\mathbf{x}_{<i})} = q(\mathcal{X}_i|\mathbf{x}_{<i})$.

*Proof.* In this proof, we make extensive use of the general Leibniz rule, which states that

$$\nabla_x^{(n)}f(x)g(x) = \sum_{m=0}^{n}\binom{n}{m}\nabla_x^{(n-m)}f(x)\nabla_x^{(m)}g(x).$$

We consider the terms $\mathbb{E}_{q_i}\left[\sum_{\mathbf{x}_k \in \mathcal{X}_k} \overrightarrow{\nabla_N^{(n)}W_i\boxdot(L_{i-1})a_i}\right]$ and the term $\mathbb{E}_{q_k}\left[\sum_{\mathbf{x}_k \in \mathcal{X}_k} \overrightarrow{\nabla_N^{(n)}W_k\boxdot(L_k)F}\right]$ separately, starting with the first.

---

[2] In other words, all deterministic functions, and all probability measures associated with the stochastic nodes are identical under evaluation.

*Lemma* 1.1. For any positive number $1 \leq j \leq k$,

$$\mathbb{E}_{q_1}\left[\sum_{\mathbf{x}_1 \in \mathcal{X}_1} \ldots \mathbb{E}_{q_j}\left[\sum_{\mathbf{x}_j \in \mathcal{X}_j} \overrightarrow{\nabla_N^{(n)} W_j \boxdot (L_j) a_j}\right] \ldots\right] = 0.$$

*Proof.* We will prove the lemma using induction. First, let $j = 1$. Then, using condition 2,

$$\mathbb{E}_{q_1}\left[\sum_{\mathbf{x}_1 \in \mathcal{X}_1} \overrightarrow{\nabla_N^{(n)} w_1 a_1}\right] = 0$$

Next, assume the inductive hypothesis holds for $j$, and consider the inner expectation of $j + 1$:

$$=\mathbb{E}_{q_{j+1}}\left[\sum_{\mathbf{x}_{j+1} \in \mathcal{X}_{j+1}} \overrightarrow{\nabla_N^{(n)} \boxdot (L_j) a_{j+1} W_{j+1}}\right] = \mathbb{E}_{q_{j+1}}\left[\sum_{\mathbf{x}_{j+1} \in \mathcal{X}_{j+1}} \overrightarrow{\nabla_N^{(n)} w_{j+1} a_{j+1} \boxdot (L_j) W_j}\right]$$

$$=\mathbb{E}_{q_{j+1}}\left[\sum_{\mathbf{x}_{j+1} \in \mathcal{X}_{j+1}} \sum_{m=0}^{n} \overrightarrow{\binom{n}{m} \nabla_N^{(m)} w_{j+1} a_{j+1} \nabla_N^{(n-m)} \boxdot (L_j) W_j}\right]$$

Next, note that $W_j$ and $A_j$ are both independent of $\mathbf{x}_{j+1}$. Therefore, they can be moved out of the expectation. To do this, we implicitly use condition 4 to move the $\rightarrow$ operator through the expectation.

$$\mathbb{E}_{q_{j+1}}\left[\sum_{\mathbf{x}_{j+1} \in \mathcal{X}_{j+1}} \sum_{m=0}^{n} \binom{n}{m} \overrightarrow{\nabla_N^{(m)} w_{j+1} a_{j+1} \nabla_N^{(n-m)} \boxdot (L_j) W_j}\right]$$

$$= \sum_{m=0}^{n} \binom{n}{m} \overrightarrow{\nabla_N^{(n-m)} \boxdot (L_j) W_j} \mathbb{E}_{q_{j+1}}\left[\sum_{\mathbf{x}_{j+1} \in \mathcal{X}_{j+1}} \overrightarrow{\nabla_N^{(m)} w_{j+1} a_{j+1}}\right]$$

By condition 2 of the theorem, $\mathbb{E}_{q_{j+1}}\left[\sum_{\mathbf{x}_{j+1} \in \mathcal{X}_{j+1}} \overrightarrow{\nabla_N^{(m)} w_{j+1} a_{j+1}}\right] = 0$. Therefore, we can remove this term and conclude that

$$\mathbb{E}_{q_1}\left[\sum_{\mathbf{x}_1 \in \mathcal{X}_1} \ldots \mathbb{E}_{q_{j+1}}\left[\sum_{\mathbf{x}_{j+1} \in \mathcal{X}_{j+1}} \overrightarrow{\nabla_N^{(n)} W_{j+1} \boxdot (L_j) a_{j+1}}\right] \ldots\right] = 0.$$

$\square$

Next, we consider the term $\mathbb{E}_{q_k}\left[\sum_{\mathbf{x}_k \in \mathcal{X}_k} \overrightarrow{\nabla_N^{(n)} W_k \boxdot (L_k) F}\right]$ and prove using induction that

*Lemma* 1.2. For any $1 \leq j \leq k$, it holds that

$$\mathbb{E}_{q_1}\left[\sum_{\mathbf{x}_1 \in \mathcal{X}_1} \ldots \mathbb{E}_{q_j}\left[\sum_{\mathbf{x}_j \in \mathcal{X}_j} \overrightarrow{\nabla_N^{(n)} W_j \boxdot (L_j) F'}\right] \ldots\right] = \nabla_N^{(n)} \mathbb{E}[F]$$

where $F' = \mathbb{E}_{\mathcal{S}_{j+1}, \ldots, \mathcal{S}_k}[F]$. Furthermore, for $1 < j \leq k$, it holds that

$$\mathbb{E}_{q_1}\left[\sum_{\mathbf{x}_1 \in \mathcal{X}_1} \ldots \mathbb{E}_{q_j}\left[\sum_{\mathbf{x}_j \in \mathcal{X}_j} \overrightarrow{\nabla_N^{(n)} W_j \boxdot (L_j) F'}\right] \ldots\right]$$

$$=\mathbb{E}_{q_1}\left[\sum_{\mathbf{x}_1 \in \mathcal{X}_1} \ldots \mathbb{E}_{q_{j-1}}\left[\sum_{\mathbf{x}_{j-1} \in \mathcal{X}_{j-1}} \overrightarrow{\nabla_N^{(n)} W_{j-1} \boxdot (L_{j-1}) \mathbb{E}_{\mathcal{S}_j}[F']}\right] \ldots\right]$$

*Proof.* The base case $j = 1$ directly follows from condition 1:

$$\mathbb{E}_{q_1}\left[\sum_{\mathbf{x}_1 \in \mathcal{X}_1} \overrightarrow{\nabla_N^{(n)} w_1 \boxdot (l_i) F'}\right] = \overrightarrow{\nabla_N^{(n)} \mathbb{E}_{\mathcal{S}_1}[F']} = \nabla_N^{(n)} \mathbb{E}[F],$$

since $\mathbb{E}[F] = \mathbb{E}_{\mathcal{S}_1, \ldots, \mathcal{S}_k}[F]$ and by the assumption that $\mathbb{E}[F]$ is identical under evaluation.

Assume the lemma holds for $j < k$ and consider $j+1$. First, we use Proposition 4 and reorder the terms:

$$\mathbb{E}_{q_{j+1}}\left[\sum_{\mathbf{x}_{j+1}\in\mathcal{X}_{j+1}}\overrightarrow{\nabla_N^{(n)}W_{j+1}\square(L_{j+1})F'}\right]$$

$$=\mathbb{E}_{q_{j+1}}\left[\sum_{\mathbf{x}_j\in\mathcal{X}_j}\overrightarrow{\nabla_N^{(n)}W_{j+1}\square(L_{j+1})w_{j+1}\square(l_{j+1})F'}\right]$$

Next, we again use the general Leibniz rule:

$$\mathbb{E}_{q_{j+1}}\left[\sum_{\mathbf{x}_{j+1}\in\mathcal{X}_{j+1}}\overrightarrow{\nabla_N^{(n)}W_j\square(L_j)w_{j+1}\square(l_{j+1})F'}\right]$$

$$=\mathbb{E}_{q_{j+1}}\left[\sum_{\mathbf{x}_{j+1}\in\mathcal{X}_{j+1}}\sum_{m=0}^{n}\overrightarrow{\binom{n}{m}\nabla_N^{(n-m)}W_j\square(L_j)\nabla_N^{(m)}w_{j+1}\square(l_{j+1})F'}\right]$$

where we use for the general Leibniz rule $f = W_j\square(L_j)$ and $g = w_{j+1}\square(l_{j+1})F'$. Note that $\nabla_N^{(n-m)}W_j\square(L_j)$ does not depend on $\mathbf{x}_{j+1}$. Therefore,

$$\mathbb{E}_{q_{j+1}}\left[\sum_{\mathbf{x}_{j+1}\in\mathcal{X}_{j+1}}\sum_{m=0}^{n}\overrightarrow{\binom{n}{m}\nabla_N^{(n-m)}W_j\square(L_j)\nabla_N^{(m)}w_{j+1}\square(l_{j+1})F'}\right]$$

$$=\sum_{m=0}^{n}\binom{n}{m}\overrightarrow{\nabla_N^{(n-m)}W_j\square(L_j)}\,\mathbb{E}_{q_{j+1}}\left[\sum_{\mathbf{x}_{j+1}\in\mathcal{X}_{j+1}}\overrightarrow{\nabla_N^{(m)}w_{j+1}\square(l_{j+1})F'}\right]$$

$$=\sum_{m=0}^{n}\binom{n}{m}\overrightarrow{\nabla_N^{(n-m)}W_j\square(L_j)\nabla_N^{(m)}\mathbb{E}_{\mathcal{S}_{j+1}}[F']}$$

$$=\overrightarrow{\nabla_N^{(n)}W_j\square(L_j)\mathbb{E}_{\mathcal{S}_{j+1}}[F']}$$

From lines 2 to 3, we use condition 1 to reduce the expectation. In the last line, we use the general Leibniz rule in the other direction. We showed that

$$\mathbb{E}_{q_1}\left[\sum_{\mathbf{x}_1\in\mathcal{X}_1}\ldots\mathbb{E}_{q_{j+1}}\left[\sum_{\mathbf{x}_{j+1}\in\mathcal{X}_{j+1}}\overrightarrow{\nabla_N^{(n)}W_{j+1}\square(L_{j+1})F'}\right]\ldots\right]$$

$$=\mathbb{E}_{q_1}\left[\sum_{\mathbf{x}_1\in\mathcal{X}_1}\ldots\mathbb{E}_{q_j}\left[\sum_{\mathbf{x}_j\in\mathcal{X}_j}\overrightarrow{\nabla_N^{(n)}W_j\square(L_j)\mathbb{E}_{\mathcal{S}_{j+1}}[F']}\right]\ldots\right]=\nabla_N^{(n)}\mathbb{E}[F]$$

where we use the inductive hypothesis from step 2 to 3, using that $\mathbb{E}_{\mathcal{S}_{j+1}}[F'] = \mathbb{E}_{\mathcal{S}_{j+1},\ldots,\mathcal{S}_k}[F]$. $\square$

Using these two lemmas and condition 4, it is easy to show the theorem:

$$\mathbb{E}_{q_1}\left[\sum_{\mathbf{x}_1\in\mathcal{X}_1}\ldots\mathbb{E}_{q_k}\left[\sum_{\mathbf{x}_k\in\mathcal{X}_k}\overrightarrow{\nabla_N^{(n)}W_k\Big(A_k+\square(L_k)F\Big)}\right]\ldots\right]$$

$$=\mathbb{E}_{q_1}\left[\sum_{\mathbf{x}_1\in\mathcal{X}_1}\ldots\mathbb{E}_{q_k}\left[\sum_{\mathbf{x}_k\in\mathcal{X}_k}\overrightarrow{\nabla_N^{(n)}W_k A_k}\right]\ldots\right]$$

$$+\mathbb{E}_{q_1}\left[\sum_{\mathbf{x}_1\in\mathcal{X}_1}\ldots\mathbb{E}_{q_k}\left[\sum_{\mathbf{x}_k\in\mathcal{X}_k}\overrightarrow{\nabla_N^{(n)}W_k\square(L_k)F}\right]\ldots\right]$$

$$=0+\nabla_N^{(n)}\mathbb{E}[F]=\nabla_N^{(n)}\mathbb{E}[F]$$

$\square$

Note that we used in the proof that condition 1 implies that $\nabla_N^{(n)}\mathbb{E}_{q_i}[\sum_{\mathbf{x}_i\in\mathcal{X}_i}\overrightarrow{w_i\square(l_i)}] = 0$, which can be seen by taking $f(\mathbf{x}_i) = 1$ and noting that $\nabla_N^{(m)}\mathbb{E}_{\mathcal{S}_i}[1] = 0$ for $n > 0$.

# C  Any-order control variate

Many gradient estimators are combined with control variates to reduce variance. We consider control variates for any-order derivative estimation. [33] introduces an any-order baseline in the context of score functions, but only provides proof that this is the baseline for the second-order gradient estimate. We use the *Storchastic* framework to prove that it is also the correct baseline for any-order derivatives[3]. Furthermore, we generalize the ideas behind this baseline to all control variates, instead of just score-function baselines.

The control variate that implements any-order baselines is:

$$a_i(\mathbf{x}_{<i}, \mathcal{X}_i) = (1 - \boxdot(l_i))b_i(\mathbf{x}_{<i}, \mathcal{X}_i \setminus \{x_i\}).$$

First, we show that baselines satisfy condition 2 of Theorem 1. We will assume here that we take only 1 sample with replacement, but the result generalizes to taking multiple samples in the same way as for the first-order baseline. For $n = 0$, the any-order baseline evaluates to zero which can be seen by considering $\overrightarrow{1 - \boxdot(l_i)}$. If $n > 0$, then noting that $b_i$ is independent of $x_i$,

$$\mathbb{E}_{q_i}[\nabla_N^{(n)}\overrightarrow{(1 - \boxdot(l_i))b_i}] = \mathbb{E}_{x_i}[\overrightarrow{-b_i\nabla_N^{(n)} - \boxdot(l_i)}] = \overrightarrow{-b_i}\nabla_N^{(n)}\mathbb{E}_{x_i}[1] = 0$$

We next provide a proof for the validity of this baseline for variance reduction of any-order gradient estimation. To do this, we first prove a new general result on the $\boxdot$ operator:

**Proposition 7.** *For any sequence of functions* $\{l_1, ..., l_k\}$, $\boxdot(L_k)$ *is equivalent under evaluation for orders of differentiation* $n > 0$ *to* $\sum_{i=1}^{k}(\boxdot(l_i) - 1)\boxdot(L_{i-1})$. *That is, for all positive numbers* $n > 0$,

$$\overrightarrow{\nabla_N^{(n)}\boxdot(L_k)} = \nabla_N^{(n)}\overrightarrow{\sum_{i=1}^{k}\left(\boxdot(l_i) - 1\right)\boxdot(L_{i-1})}$$

*Proof.* We will prove this using induction on $k$, starting with the base case $k = 1$. Since $n > 0$,

$$\overrightarrow{\nabla_N^{(n)}(\boxdot(l_1) - 1)\boxdot(0)} = \overrightarrow{\boxdot(0)\nabla_N^{(n)}\boxdot(l_1)} = \overrightarrow{\nabla_N^{(n)}\boxdot(l_1)}$$

Next, assume the proposition holds for $k$ and consider $k + 1$. Then by splitting up the sum,

$$\nabla_N^{(n)}\overrightarrow{\sum_{i=1}^{k+1}(\boxdot(l_i) - 1)\boxdot(L_{i-1})} = \nabla_N^{(n)}\overrightarrow{(\boxdot(l_{k+1}) - 1)\boxdot(L_k)} + \nabla_N^{(n)}\overrightarrow{\sum_{i=1}^{k}(\boxdot(l_i) - 1)\boxdot(L_{i-1})}$$

$$= \nabla_N^{(n)}\overrightarrow{(\boxdot(l_{k+1}) - 1)\boxdot(L_k)} + \nabla_N^{(n)}\overrightarrow{\boxdot(L_k)}$$

where in the second step we use the inductive hypothesis.

We will next consider the first term using the general Leibniz rule:

$$\nabla_N^{(n)}\overrightarrow{(\boxdot(l_{k+1}) - 1)\boxdot(L_k)} = \sum_{m=0}^{n}\binom{n}{m}\nabla_N^{(m)}\overrightarrow{(\boxdot(l_{k+1}) - 1)}\nabla_N^{(n-m)}\overrightarrow{\boxdot(L_k)}$$

We note that the term corresponding to $m = 0$ can be ignored, as $\overrightarrow{\boxdot(l_{k+1}) - 1} = (1 - 1) = 0$. Furthermore, for $m > 0$, $\nabla_N^{(m)}\overrightarrow{(\boxdot(l_{k+1}) - 1)} = \nabla_N^{(m)}\overrightarrow{\boxdot(l_{k+1})}$. Therefore,

$$\nabla_N^{(n)}\overrightarrow{(\boxdot(l_{k+1}) - 1)\boxdot(L_k)} = \sum_{m=1}^{n}\binom{n}{m}\nabla_N^{(m)}\overrightarrow{\boxdot(l_{k+1})}\nabla_N^{(n-m)}\boxdot(L_k)$$

---

[3]We use a slight variant of the baseline introduced in [33] to solve an edge case. We will explain in the end of this section how they differ.

Finally, we add the other term $\nabla_N^{(n)}\boxdot(L_k)$ again. Then using the general Leibniz rule in the other direction and Proposition 4,

$$\overrightarrow{=\sum_{m=1}^{n}\binom{n}{m}\nabla_N^{(m)}\boxdot(l_{k+1})\nabla_N^{(n-m)}\boxdot(L_k)+\nabla_N^{(n)}\boxdot(L_k)}$$

$$\overrightarrow{=\sum_{m=1}^{n}\binom{n}{m}\nabla_N^{(m)}\boxdot(l_{k+1})\nabla_N^{(n-m)}\boxdot(L_k)+\boxdot(l_{k+1})\nabla_N^{(n)}\boxdot(L_k)}$$

$$\overrightarrow{=\sum_{m=0}^{n}\binom{n}{m}\nabla_N^{(m)}\boxdot(l_{k+1})\nabla_N^{(n-m)}\boxdot(L_k)}=\overrightarrow{\nabla_N^{(n)}\boxdot(l_{k+1})\boxdot(L_k)}=\overrightarrow{\nabla_N^{(n)}\boxdot(L_{k+1})}$$

$\square$

Next, we note that we can rewrite the expectation of the *Storchastic* surrogate loss in Equation (3) to

$$\mathbb{E}_{q_1}\left[\sum_{\mathbf{x}_1\in\mathcal{X}_1}+\ldots\mathbb{E}_{q_k}\left[\sum_{\mathbf{x}_k\in\mathcal{X}_k}\overrightarrow{\nabla_N^{(n)}W_k\Big(A_k+\boxdot(L_k)F\Big)}\right]\ldots\right]$$

where $A_k=\sum_{i=1}^{k}\boxdot\Big(\sum_{j=1}^{i-1}l_j\Big)a_i$. This can be seen by using Condition 1 and 4 of Theorem 1 to iteratively move the $\boxdot\Big(\sum_{j=1}^{i-1}l_j\Big)a_i$ terms into the expectations, which is allowed since they don't depend on $\mathcal{S}_{>i}$.

**Theorem 2.** *Under the conditions of Theorem 1,*

$$A_k+\boxdot(L_k)F\stackrel{\Rrightarrow}{=}\sum_{i=1}^{k}\boxdot(L_{i-1})(a_i+(\boxdot(l_i)-1)F)+F,$$

*where $A_k=\sum_{i=1}^{k}\boxdot\Big(\sum_{j=1}^{i-1}l_j\Big)a_i$.*

*Proof.*

$$\overrightarrow{\nabla_N^{(n)}(A_k+\boxdot(L_k)F)}=\overrightarrow{\nabla_N^{(n)}A_k+\sum_{m=0}^{n}\binom{n}{m}\nabla_N^{(m)}\boxdot(L_k)\nabla_N^{(n-m)}F}\tag{4}$$

$$\overrightarrow{=\nabla_N^{(n)}A_k+\sum_{m=1}^{n}\binom{n}{m}\nabla_N^{(m)}\boxdot(L_k)\nabla_N^{(n-m)}F+\nabla_N^{(n)}F}\tag{5}$$

$$\overrightarrow{=\nabla_N^{(n)}A_k+\sum_{m=1}^{n}\binom{n}{m}\nabla_N^{(m)}\sum_{i=1}^{k}(\boxdot(l_i)-1)\boxdot(L_{i-1})\nabla_N^{(n-m)}F+\nabla_N^{(n)}F}\tag{6}$$

$$\overrightarrow{=\nabla_N^{(n)}\big(\sum_{i=1}^{k}\boxdot(L_{i-1})a_i+\sum_{i=1}^{k}(\boxdot(l_i)-1)\boxdot(L_{i-1})F+F\big)}\tag{7}$$

$$\overrightarrow{=\nabla_N^{(n)}\big(\sum_{i=1}^{k}\boxdot(L_{i-1})(a_i+(\boxdot(l_i)-1)F)+F\big)}$$

From (4) to (5), we use that $m=0$ evaluates to $\nabla_N^{(n)}F$. From (5) to (6), we use Proposition 7. From (6) to (7), we do a reversed general Leibniz rule on the second term. To be able do that, we use that setting $m=0$ in the second term would evaluate to 0 as $\overrightarrow{\boxdot(l_i)-1}=0$. $\square$

Next, consider the inner computation of the *Storchastic* framework in which all $a_i$ use a baseline of the form in Equation C. Note that $a_i = 0$ is also in this form by setting $b_i = 0$. Assume $n > 0$ and without loss of generality[4] assume $\nabla_N^{(m)} w_i = 0$ for all $m$ and $i$. Then using Proposition 7,

$$\prod_{i=1}^{k} w_i \nabla_N^{(n)} \overrightarrow{\Big( \sum_{i=1}^{k} (1 - \square(l_i)) \square(L_{i-1}) b_i + \square(L_k) F \Big)}$$

$$= \prod_{i=1}^{k} w_i \nabla_N^{(n)} \Big( -\sum_{i=1}^{k} (\square(l_i) - 1) \square(L_{i-1}) b_i + \overrightarrow{\sum_{i=1}^{k} (\square(l_i) - 1) \square(L_{i-1}) F} \Big)$$

$$= \prod_{i=1}^{k} w_i \nabla_N^{(n)} \overrightarrow{\sum_{i=1}^{k} (\square(l_i) - 1) \square(L_{i-1})} (F - b_i)$$

The intuition behind the variance reduction of this any-order gradient estimate is that all terms of the gradient involving $l_i$, possibly multiplied with other $l_j$ such that $j < i$, use the $i$-th baseline $b_i$. This allows modelling baselines for each sampling step to effectively make use of background knowledge or known statistics of the corresponding set of random variables.

We note that our baseline is slightly different from [33], which instead of $\square(L_{i-1}) = \square(\sum_{j=1}^{i-1} l_j)$ used $\square(\sum_{\mathcal{S}_j \prec \mathcal{S}_i} l_j)$. Although this might initially seem more intuitive, we will show with a small counterexample why we should consider any stochastic nodes ordered topologically before $i$ instead of just those that directly influence $i$.

Consider the stochastic computation graph with stochastic nodes $p(S_1|N)$ and $p(S_2|N)$ and cost function $f(x_1, x_2)$. For simplicity, assume we use single-sample score function estimators for each stochastic node. Consider the second-order gradient of the cost function using the recursion in Proposition 2:

$$\nabla_N^2 \mathbb{E}_{S_1, S_2}[f(x_1, x_2)] = \mathbb{E}_{S_1, S_2}[\nabla_N^2 \square \overrightarrow{(\sum_{i=1}^{2} \log p(x_i|N)) f(x_1, x_2)}]$$

$$= \mathbb{E}_{S_1, S_2}[f(x_1, x_2) \Big( \sum_{i=1}^{2} \nabla_N^2 \log p(x_i|N) + (\nabla_N \log p(x_i|N))^2$$

$$+ 2\nabla_N \log p(x_1|N) \nabla_N \log p(x_2|N)))\Big)]$$

Despite the fact that $x_1$ does not directly influence $x_2$, higher-order derivatives will have terms that involve both the log-probabilities of $x_1$ and $x_2$, in this case $2\nabla_N \log p(x_1|N) \nabla_N \log p(x_2|N)$. Note that since $a$ does not directly influence $b$, the baseline generated for the second-order derivative using the method in [33] would be

$$\nabla_N^2 \overrightarrow{\sum_{i=1}^{2} a_i} = \overrightarrow{\sum_{i=1}^{2} \nabla_N^2 (1 - \square(\log p(x_i|N))) \square(0) b_i} = -\sum_{i=1}^{2} \nabla_N^2 \log p(x_i|N))) b_i$$

This baseline does not have a term for $2\nabla_N \log p(x_1|N) \nabla_N \log p(x_2|N)$, meaning the variance of that term will not be reduced through a baseline. The baseline introduced in Equation C will include it, since

$$\nabla_N^2 \overrightarrow{\sum_{i=1}^{2} a_i} = \overrightarrow{\nabla_N^2 (1 - \square(\log p(x_1|N))) b_i + (1 - \square(\log p(x_2|N))) \square(\log p(x_1|N)) b_i}$$

$$= -\sum_{i=1}^{2} \nabla_N^2 \log p(x_i|N))) b_i - 2\nabla_N \log p(x_1|N) \nabla_N \log p(x_2|N)$$

---

[4]This is assumed simply to make the notation clearer. If the weights are differentiable, the same thing can be shown using an application of the general Leibniz rule.

Designing a good baseline function $b_{i,j}(\mathbf{x}_{<i,j}, \mathcal{X}_i \setminus \{x_{i,j}\})$ that will reduce variance significantly is highly application dependent. Simple options are a moving average and the leave-one-out baseline, which is given by $b_i(\mathbf{x}_{<i}, \mathcal{X}_i \setminus \{x_i\}) = \frac{1}{m-1} \sum_{j'=1, j' \neq j}^{m} \perp(\circ) f(\mathbf{x}_{<i}, x_{i,j'}))$ [22, 35]. More advanced baselines can take into account the previous stochastic nodes $\mathcal{S}_1, ..., \mathcal{S}_{i-1}$ [48]. Here, one should only consider the stochastic nodes that directly influence $S_i$, that is, $\mathcal{S}_{\prec i}$. Another popular choice is self-critical baselines [42, 21] that use deterministic test-time decoding algorithms to find $\hat{\mathbf{x}}_i$ and then evaluate it, giving $b_i(\mathbf{x}_{<i}, \mathcal{X}_i \setminus \{x_i\}) = f(\hat{\mathbf{x}}_i)$.

# D    Examples of Gradient Estimators

In this section, we prove the validity of several gradient estimators within the *Storchastic* framework, focusing primarily on discrete gradient estimation methods.

## D.1    Expectation

Assume $p(x_i)$ is a discrete (ie, categorical) distribution with a finite amount of classes $1, ..., C_i$. While this is not an estimate but the true gradient, it fits in the *Storchastic* framework as follows:

1. $w_i(x_i) = p(x_i|\mathbf{x}_{<i})$
2. $q(\mathcal{X}_i|\mathbf{x}_{<i}) = \delta_{\{1,...,C_i\}}(\mathcal{X}_i)$ (that is, a dirac delta distribution with full mass on sampling exactly the sequence $\{1, ..., C_i\}$)
3. $l_i(x_i) = 0$
4. $a_i(x_i) = 0$

Next, we prove the individual conditions to show that this method can be used within *Storchastic*, starting with condition 1:

$$\mathbb{E}_{q_i}\Big[ \sum_{x_i \in \mathcal{X}i} \overrightarrow{\nabla_N^{(n)} w_i \boxdot (l_i) f(x_i)} \Big] = \sum_{j=1}^{C_i} \overrightarrow{\nabla_N^{(n)} p(x_i = j|\mathbf{x}_{<i}) \boxdot (0) f(j)}$$

$$= \overrightarrow{\sum_{j=1}^{C_i} \sum_{m=0}^{n} \nabla_N^{(n-m)} p(x_i = j|\mathbf{x}_{<i}) \nabla_N^{(m)} \boxdot (0) f(j)}$$

Using the recursion in Proposition 2, we see that $\overrightarrow{\nabla_N^{(m)} \boxdot (0) f(j)} = \nabla_N^{(m)} f(j)$, since $\nabla_N l_i = \nabla_N 0 = 0$. So,

$$\overrightarrow{\sum_{j=1}^{C_i} \sum_{m=0}^{n} \nabla_N^{(n-m)} p(x_i = j|\mathbf{x}_{<i}) \nabla_N^{(m)} f(j)} = \overrightarrow{\sum_{j=1}^{C_i} \nabla_N^{(n)} p(x_i = j|\mathbf{x}_{<i}) f(j)} = \nabla_N^{(n)} \mathbb{E}_{x_i}[f(x_i)].$$

Condition 2 follows simply from $a_i(x_i) = 0$, and condition 3 follows from the fact that $\sum_{j=1}^{C_i} p(x_i = j|\mathbf{x}_{<i}) = 1$, that is, constant. Condition 4 follows from the SCG being identical under evaluation, ie $\overrightarrow{p(x_i = j|\mathbf{x}_{<i})} = p(x_i = j|\mathbf{x}_{<i})$.

It should be noted that this proof is not completely trivial, as it shows how to implement the expectation so that it can be combined with other gradient estimators while making sure the pathwise derivative through $f$ also gets the correct gradient.

## D.2    Score Function

The score function is the best known general gradient estimator and is easy to fit in *Storchastic*.

### D.2.1    Score Function with Replacement

We consider the case where we take $m$ samples with replacement from the distribution $p(x_i|\mathbf{x}_{<i})$, and we use a baseline $b_i(\mathbf{x}_{<i}, \mathcal{X}_i \setminus \{x_i\})$ for the first-order gradient estimate.

1. $w_i(x_i) = \frac{1}{m}$
2. $q_i = \prod_{j=1}^m p(x_{i,j}|\mathbf{x}_{<i})$. That is, $x_{i,1}, ..., x_{i,m} \sim p(x_i|\mathbf{x}_{<i})$.
3. $l_i(x_i) = \log p(x_i|\mathbf{x}_{<i})$
4. $a_i(\mathbf{x}_{<i}, \mathcal{X}_i) = (1-\boxdot(l_i))b_i(\mathbf{x}_{<i}, \mathcal{X}_i \setminus \{x_i\})$, where $b_i(\mathbf{x}_{<i}, \mathcal{X}_i \setminus \{x_i\})$ is not differentiable, that is, $\nabla_N^{(n)} b_i(\mathbf{x}_{<i}, \mathcal{X}_i \setminus \{x_i\}) = 0$ for $n > 0$.

We start by showing that condition 1 holds. We assume $p(x_i|\mathbf{x}_{<i})$ is a continuous distribution and note that the proof for discrete distributions is analogous.

We will show how to prove that sampling a set of $m$ samples with replacement can be reduced in expectation to sampling a single sample. Here, we use that $x_{i,1}, ..., x_{i,m}$ are all independently (line 1 to 2) and identically (line 2 to 3) distributed.

$$\mathbb{E}_{q_i}[\sum_{j=1}^m \overrightarrow{\nabla_N^{(n)} \frac{1}{m}\boxdot(l_{i,j})f(\mathbf{x}_{<i}, x_{i,j})}] = \frac{1}{m}\sum_{j=1}^m \mathbb{E}_{x_{i,j}\sim p(x_i)}[\overrightarrow{\nabla_N^{(n)}\boxdot(l_{i,j})f(\mathbf{x}_{\le i}, x_{i,j})}]$$

$$= \frac{1}{m}\sum_{j=1}^m \mathbb{E}_{x_i\sim p(x_i)}[\overrightarrow{\nabla_N^{(n)}\boxdot(l_i)f(\mathbf{x}_{\le i})}] = \mathbb{E}_{x_i}\left[\overrightarrow{\nabla_N^{(n)}\boxdot(l_i)f(\mathbf{x}_{\le i})}\right]$$

A proof that $\mathbb{E}_{x_i}\left[\overrightarrow{\nabla_N^{(n)}\boxdot(l_i)f(\mathbf{x}_{\le i})}\right] = \nabla_N^{(n)}\mathbb{E}_{x_i}[f(\mathbf{x}_{\le i})]$ was first given in [13]. For completeness, we give a similar proof here, using induction.

First, assume $n = 0$. Then, $\mathbb{E}_{x_i}[\overrightarrow{\boxdot(l_i)}f(\mathbf{x}_{\le i})] = \mathbb{E}_{x_i}[\overrightarrow{f(\mathbf{x}_{\le i})}] = \mathbb{E}_{x_i}[f(\mathbf{x}_{\le i}]$.

Next, assume it holds for $n$, and consider $n + 1$. Using Proposition 2, we find that $g^{(n+1)}(\mathbf{x}_{\le i}) = \nabla_N g^{(n)}(\mathbf{x}_{\le i}) + g^{(n)}(\mathbf{x}_{\le i})\nabla_N \log p(x_i|\mathbf{x}_{<i})$. Writing the expectation out, we find

$$\mathbb{E}_{x_i}[\overrightarrow{\nabla_N g^{(n)}(\mathbf{x}_{\le i}) + g^{(n)}(\mathbf{x}_{\le i})\nabla_N \log p(x_i|\mathbf{x}_{<i})}]$$

$$= \int \overrightarrow{p(x_i|\mathbf{x}_{<i})(\nabla_N g^{(n)}(\mathbf{x}_{\le i}) + g^{(n)}(\mathbf{x}_{\le i})\frac{\nabla_N p(x_i|\mathbf{x}_{<i})}{p(x_i|\mathbf{x}_{<i})})}dx_i$$

$$= \int \overrightarrow{\nabla_N p(x_i|\mathbf{x}_{<i})g^{(n)}(\mathbf{x}_{\le i})}dx_i = \overrightarrow{\nabla_N \mathbb{E}_{x_i}[g^{(n)}(\mathbf{x}_{\le i})]}$$

By Proposition 3, $g^{(n)}(x_i)$ is identical under evaluation, since by the assumption of Theorem 1 both $p(x_i|\mathbf{x}_{<i})$ and $f(\mathbf{x}_{\le i})$ are identical under evaluation. As a result, $\overrightarrow{\nabla_N \mathbb{E}_{x_i}[g^{(n)}(\mathbf{x}_{\le i})]} = \nabla_N \mathbb{E}_{x_i}[\overrightarrow{g^{(n)}(\mathbf{x}_{\le i})}]$. Therefore, by the induction hypothesis,

$$\mathbb{E}_{q_i}[\overrightarrow{g^{(n+1)}(\mathbf{x}_{\le i})}] = \nabla_N \mathbb{E}_{x_i}[\overrightarrow{g^{(n)}(\mathbf{x}_{\le i})}] = \nabla_N^{(n+1)}\mathbb{E}_{x_i}[f(\mathbf{x}_{\le i})]$$

Since the weights ($\frac{1}{m}$) are constant, condition 3 is satisfied.

### D.2.2 Importance Sampling

A common use case for weighting samples is importance sampling [44]. In the context of gradient estimation, it is often used in off-policy reinforcement-learning [32] to allow unbiased gradient estimates using samples from another policy. For simplicity, we consider importance samples within the context of score function estimators, single-sample estimates, and use no baselines. The last two can be introduced using the techniques in Section D.2.1 and C.

1. $w_i = \perp(\frac{p(x_i|\mathbf{x}_{<i})}{q(x_i|\mathbf{x}_{<i})})$,
2. $q(x_i|\mathbf{x}_{<i})$ is the sampling distribution,
3. $l_i(\mathbf{x}_i) = \log p(x_i|\mathbf{x}_{<i})$,
4. $a_i(\mathbf{x}_{<i}, \mathcal{X}_i) = 0$.

Condition 3 follows from the fact that $\nabla_N^{(n)} w_i = 0$ for $n > 0$, since the importance weights are detached from the computation graph. Condition 1:

$$\mathbb{E}_{q_i}[\overrightarrow{\nabla_N^{(n)} \bot \Big(\frac{p(x_i|\mathbf{x}_{<i})}{q(x_i|\mathbf{x}_{<i})}\Big)\boxdot(l_i)f(\mathbf{x}_{\leq i})}] = \int_{\Omega_i} q(x_i|\mathbf{x}_{<i})\frac{p(x_i|\mathbf{x}_{<i})}{q(x_i|\mathbf{x}_{<i})}\overrightarrow{\nabla_N^{(n)}\boxdot(l_i)f(\mathbf{x}_{\leq i})}dx_i$$

$$=\mathbb{E}_{\mathcal{S}_i}[\overrightarrow{\nabla_N^{(n)}\boxdot(l_i)f(\mathbf{x}_{\leq i})}] = \overrightarrow{\mathbb{E}_{\mathcal{S}_i}[f(\mathbf{x}_{\leq i})]}$$

where in the last step we use the proven condition 1 of D.2.1. Note that this holds both for $n = 0$ and $n > 0$.

### D.2.3 Discrete Sequence Estimators

Recent literature introduced several estimators for sequences of discrete random variables. These are quite similar in how they are implemented in *Storchastic*, which is why we group them together.

The sum-and-sample estimator chooses a set of sequences $\hat{\mathcal{X}}_i \subset \Omega_i$ and chooses $k - |\hat{\mathcal{X}}_i| > 0$ samples from $\Omega_i \setminus \hat{\mathcal{X}}_i$. This set can be the most probable sequences [30] or can be chosen randomly [25]. This is guaranteed not to increase variance through Rao-Blackwellization [6, 30]. It is often used together with deterministic cost functions $f$, which allows memorizing the cost-function evaluations of the sequences in $\hat{\mathcal{X}}_i$. In this context, the estimator is known as Memory-Augmented Policy Optimization [28].

1. $w_i(\mathbf{x}_i) = I[\mathbf{x}_i \in \hat{\mathcal{X}}_i]p(\mathbf{x}_i|\mathbf{x}_{<i}) + I[\mathbf{x}_i \notin \hat{\mathcal{X}}_i]\frac{p(\mathbf{x}_i \notin \hat{\mathcal{X}}_i)}{k-|\hat{\mathcal{X}}_i|}$

2. $q(\mathcal{X}_i) = \delta_{\hat{\mathcal{X}}_i}(\mathbf{x}_{i,1}, ..., \mathbf{x}_{i,|\hat{\mathcal{X}}_i|}) \cdot \prod_{j=|\hat{\mathcal{X}}_i|+1}^{k} p(\mathbf{x}_{i,j}|\mathbf{x}_{i,j} \notin \hat{\mathcal{X}}_i, \mathbf{x}_{<i})$

were $p(\mathbf{x}_i \notin \hat{\mathcal{X}}_i) = 1 - \sum_{\mathbf{x}_{i'} \in \hat{\mathcal{X}}_i} p(\mathbf{x}_{i'}|\mathbf{x}_{<i})$. This essentially always 'samples' the set $\hat{\mathcal{X}}_i$ using the Dirac delta distribution, and then samples $k$ more samples out of the remaining sequences, with replacement. The estimator resulting from this implementation is

$$\mathbb{E}_{q_i}[\sum_{j=1}^{|\hat{\mathcal{X}}_i|} p(\mathbf{x}_{i,j}|\mathbf{x}_{<i})f(\mathbf{x}_{<i}, \mathbf{x}_{i,j}) + \sum_{j=|\hat{\mathcal{X}}_i|+1}^{k} \frac{p(\mathbf{x}_i \notin \hat{\mathcal{X}}_i)}{k-|\hat{\mathcal{X}}_i|}\boxdot(l_i)f(\mathbf{x}_{<i}, \mathbf{x}_{i,j})]$$

Using the result from Section D.1, we see that

$$\overrightarrow{\nabla_N^{(n)}\mathbb{E}_{q_i}[\sum_{j=1}^{|\hat{\mathcal{X}}_i|} p(\mathbf{x}_{i,j}|\mathbf{x}_{<i})f(\mathbf{x}_{<i}, \mathbf{x}_{i,j})]} = \overrightarrow{\nabla_N^{(n)}p(\mathbf{x}_i \in \hat{\mathcal{X}}_i)\mathbb{E}_{q_i}[\sum_{j=1}^{|\hat{\mathcal{X}}_i|} p(\mathbf{x}_{i,j}|\mathbf{x}_{i,j} \in \hat{\mathcal{X}}_i, \mathbf{x}_{<i})f(\mathbf{x}_{<i}, \mathbf{x}_{i,j})]}$$

$$=\nabla_N^{(n)}p(\mathbf{x}_i \in \hat{\mathcal{X}}_i)\mathbb{E}_{p(\mathbf{x}_i|\mathbf{x}_i \in \hat{\mathcal{X}}_i, \mathbf{x}_{<i})}[f(\mathbf{x}_{\leq i})].$$

Similarly, from the result for sampling with replacement of score functions in Section D.2.1,

$$\overrightarrow{\nabla_N^n\mathbb{E}_{q_i}[\sum_{j=|\hat{\mathcal{X}}_i|+1}^{k} \frac{p(\mathbf{x}_i \notin \hat{\mathcal{X}}_i)}{k-|\hat{\mathcal{X}}_i|}\boxdot(l_i)f(\mathbf{x}_{<i}, \mathbf{x}_{i,j})]} = \nabla_N^{(n)}p(\mathbf{x}_i \notin \hat{\mathcal{X}}_i)\mathbb{E}_{p(\mathbf{x}_i|\mathbf{x}_i \notin \hat{\mathcal{X}}_i, \mathbf{x}_{<i})}[f(\mathbf{x}_{\leq i})]$$

Added together, these form $\nabla_N^{(n)}\mathbb{E}_{\mathcal{S}_i}[f(\mathbf{x}_{\leq i})]$, which shows that the sum-and-sample estimator with the score function is unbiased for any-order gradient estimation. The variance of this estimator can be further reduced using a baseline from Section C, such as the leave-one-out baseline.

The *unordered set estimator* is a low-variance gradient estimation method for a sequence of discrete random variables $\mathcal{S}_i$ [25]. It makes use of samples without replacement to ensure that each sequence in the sampled batch will be different. We show here how to implement this estimator within *Storchastic*, leaving the proof for validity of the estimator for [25].

1. $q(\mathcal{X}_i|\mathbf{x}_{<i})$ is an ordered sample without replacement from $p(\mathcal{S}_i|\mathbf{x}_{<i})$. For sequences, samples can efficiently be taken in parallel using ancestral gumbel-top-k sampling [24, 23]. An ordered sample without replacement means that we take a sequence of samples, where the $i$th sample cannot equal the $i - 1$ samples before it.

2. $w_i(\mathbf{x}_i) = \perp\left(\frac{p(\mathbf{x}_i|\mathbf{x}_{<i})p(U=\mathcal{X}_i|o_1=\mathbf{x}_i,\mathbf{x}_{<i})}{p(U=\mathcal{X}_i|\mathbf{x}_{<i})}\right)$, where $p(U=\mathcal{X}_i|\mathbf{x}_{<i})$ is the probability of the *unorderd* sample without replacement, and $p(U=\mathcal{X}_i|o_1=\mathbf{x}_i,\mathbf{x}_{<i})$ is the probability of the unordered sample without replacement, given that, if we were to order the sample, the first of those ordered samples is $\mathbf{x}_i$.

3. $l_i(\mathbf{x}_i) = \log p(\mathbf{x}_i|\mathbf{x}_{<i})$

4. $a_i(\mathbf{x}_{<i}, \mathcal{X}_i) = (1 - \boxdot(l_i))b_i(\mathbf{x}_{<i}, \mathcal{X}_i)$, where $b_i(\mathbf{x}_{<i}, \mathcal{X}_i) =$
$\sum_{\mathbf{x}_i' \in \mathcal{X}_i} \perp\left(\frac{p(\mathbf{x}_i'|\mathbf{x}_{<i})p(U=\mathcal{X}_i|o_1=\mathbf{x}_i,o_2=\mathbf{x}_i',\mathbf{x}_{<i})}{p(U=\mathcal{X}_i|o_1=\mathbf{x}_i,\mathbf{x}_{<i})}f(\mathbf{x}_i')\right)$

This estimator essentially reweights each sample without replacement to ensure it remains unbiased under this sampling strategy. This estimator can be used for any-order differentiation, since $\mathbb{E}_{q_i}[\sum_{\mathbf{x}_i \in \mathcal{X}_i} \overrightarrow{w_i f(\mathbf{x}_i)}] = \mathbb{E}_{\mathcal{S}_i}[\overrightarrow{f(\mathbf{x}_i)}]$ (see [25] for the proof) and $\overrightarrow{\nabla_N^{(n)}w_i} = 0$ for $n > 0$. The baseline is 0 in expectation for the zeroth and first order evaluation [25]. We leave for future work whether it is also a mean-zero baseline for $n > 1$.

### D.2.4 LAX, RELAX and REBAR

REBAR [47] and LAX and RELAX [16] are single-sample score-function based methods that learn a control variate to minimize variance. The control variate is implemented using reparameterization. We start with LAX as it is simplest, and then extend the argument to RELAX, since REBAR is a special case of RELAX. We use $b_{i,\phi}$ to denote the learnable control variate. We have to assume there is no pathwise dependency of $N$ with respect to $b_{i,\phi}$. Furthermore, we assume $\mathbf{x}_i$ is a reparameterized sample of $p(\mathbf{x}_i|\mathbf{x}_{\leq i})$. The control variate component then is:

$$a_i(\mathbf{x}_{<i}, \mathcal{X}_i) = b_{i,\phi}(\mathbf{x}_{\leq i}) - \boxdot(l_i)\perp(b_{i,\phi}(\mathbf{x}_{\leq i}))$$

Since LAX uses normal single-sample score-function, we only have to show condition 2, namely that this control variate component has 0 expectation for all orders of differentiation.

$$\mathbb{E}_{\mathcal{S}_i}\left[\overrightarrow{\nabla_N^{(n)}\left(b_{i,\phi} - \boxdot(l_i)\perp(b_{i,\phi})\right)}\right] = 0$$

$\overrightarrow{\mathbb{E}_{\mathcal{S}_i}[\nabla_N^{(m)}b_{i,\phi}]}$ is the reparameterization estimate of $\overrightarrow{\nabla_N^{(m)}\mathbb{E}_{\mathcal{S}_i}[b_{i,\phi}]}$ and $\overrightarrow{\mathbb{E}_{\mathcal{S}_i}[\nabla_N^{(m)}\boxdot(\log p(\mathbf{x}_i|\mathbf{x}_{\leq i}))\perp(b_{i,\phi})]}$ is the score-function estimate under the assumption that $b_{i,\phi}$ has no pathwise dependency. As both are unbiased expectations of the $m$-th order derivative, their difference has to be 0 in expectation, proving condition 2. Furthermore, the 0th order evaluation is exactly 0. The parameters $\phi$ are trained to minimize the gradient estimate variance.

The control variate for RELAX [16], an extension of LAX to discrete random variables, is similar. It first samples a continuously relaxed input $q(z_i|\mathbf{x}_{<i})$, which is then transformed to a discrete sample $\mathbf{x}_i \sim p(\mathbf{x}_i|\mathbf{x}_{<i})$. See [16, 47] for details on how this relaxed sampling works. It also samples a relaxed input *condition on the discrete sample*, ie $q(\tilde{z}_i|\mathbf{x}_{\leq i})$. The corresponding control variate is

$$a_i(\mathbf{x}_{<i}, \mathcal{X}_i) = b_{i,\phi}(z_i) - \perp(b_{i,\phi}(z_i)) - b_{i,\phi}(\tilde{z}_i) + (2 - \boxdot(l_i))\perp(b_{i,\phi}(\tilde{z}_i))$$

Here, we subtract $\perp(b_{i,\phi}(z_i))$ to ensure the first two terms together sum to 0 during 0th order evaluation, and add $2\perp(b_{i,\phi}(\tilde{z}_i))$ to ensure the last two terms sum to 0. Note that for $n > 0$, $\overrightarrow{\nabla_N^{(n)}a_i(\mathbf{x}_{<i}, \mathcal{X}_i)} = \overrightarrow{\nabla_N^{(n)}\left(b_{i,\phi}(z_i) - b_{i,\phi}(\tilde{z}_i) - \boxdot(l_i)\perp(b_{i,\phi}(\tilde{z}_i))\right)}$. We refer the reader to [16, 47] for details on why this control variate is zero in expectation for 1st order differentiation. We note that the results extend to higher-order differentiation since the $n$-th order derivative of $\boxdot(l_i)$ gives $n$th-order score functions which are unbiased expectations of the $n$-th order derivative.

### D.2.5 ARM

ARM is a score-function based estimator for multivariate Bernouilli random variables. For our implementation, we use the baseline formulation mentioned in [51], and we follow the derivation in terms of the Logistic random variables from [9]. ARM assumes a real-valued parameter vector $\alpha$, which can be the output of a neural network. The probabilities of the Bernoulli random variable are then assumed to be $\sigma(\alpha)$ where $\sigma$ is the sigmoid function.

1. $q(\mathcal{X}_i|\mathbf{x}_{<i})$ is a reparameterized sample from the multivariate Bernouilli distribution. First, it samples $\boldsymbol{\epsilon} \sim \text{Logistic}(\mathbf{0}, \mathbf{1})$. Define $\mathbf{z}_i = \alpha + \boldsymbol{\epsilon}$ and $\tilde{\mathbf{z}}_i = \alpha - \boldsymbol{\epsilon}$. We find $\mathbf{x}_i = I[\mathbf{z}_i > \mathbf{0}]$. Then, with this procedure, $\mathbf{x}_i \sim \text{Bernouilli}(\sigma(\alpha))$.

2. $w_i(\mathbf{x}_i) = 1$

3. $l_i(\mathbf{x}_i) = \log q_\alpha(\mathbf{z}_i)$, where $q_\alpha$ is the density function of $\text{Logistic}(\alpha, 1)$.

4. $a_i(\mathbf{x}_{<i}, \mathcal{X}_i) = \boxdot(1 - l_i(\mathbf{x}_i))\frac{1}{2}(f(\mathbf{x}_{<i}, \mathbf{z}_i > \mathbf{0}) + f(\mathbf{x}_{<i}, \tilde{\mathbf{z}}_i > \mathbf{0}))$

Since $\mathbb{E}_{\mathbf{x}_i \sim \text{Bernouilli}(\sigma(\alpha_{\boldsymbol{\theta}}))}[f(\mathbf{x}_i)] = \mathbb{E}_{\boldsymbol{\epsilon} \sim \text{Logistic}(0,1)}[f(\alpha_{\boldsymbol{\theta}} + \boldsymbol{\epsilon} > \mathbf{0})] = \mathbb{E}_{\mathbf{z}_i \sim \text{Logistic}(\alpha_{\boldsymbol{\theta}}, 1)}[f(\mathbf{z}_i > \mathbf{0})]$, any unbiased estimate of the logistic reparameterization must also be an unbiased estimate of the original Bernouilli formulation. This equality follows because the CDF of the logistic distribution is the logistic function (that is, the sigmoid function). $l_i(\mathbf{x}_i)$ is the (unbiased) score function of the logistic reparameterization, which we proved to be an unbiased estimate.

The control variate has expectation 0 for zeroth and first order differentiation. This is because it relies on the score function being an odd function [5], that is, $\nabla_N \log q_\alpha(\mathbf{z}_i) = -\nabla_N \log q_\alpha(\tilde{\mathbf{z}}_i)$. Therefore, $\mathbb{E}_{\boldsymbol{\epsilon}}[(f(\mathbf{x}_{<i}, \mathbf{z}_i > 0) + f(\mathbf{x}_{<i}, \tilde{\mathbf{z}}_i > 0))\nabla_N \log q_\alpha(\mathbf{z}_i)] = \mathbb{E}_{\boldsymbol{\epsilon}}[f(\mathbf{x}_{<i}, \mathbf{z}_i > 0)\nabla_N \log q_\alpha(\mathbf{z}_i) - f(\mathbf{x}_{<i}, \tilde{\mathbf{z}}_i > 0)\nabla_N \log q_\alpha(\tilde{\mathbf{z}}_i)]$. Note that, by symmetry of the logistic distribution, $\mathbb{E}_{\boldsymbol{\epsilon}}[f(\mathbf{x}_{<i}, \mathbf{z}_i > 0)\nabla_N \log q_\alpha(\mathbf{z}_i)] = -\mathbb{E}_{\boldsymbol{\epsilon}}[f(\mathbf{x}_{<i}, \tilde{\mathbf{z}}_i > 0)\nabla_N \log q_\alpha(\tilde{\mathbf{z}}_i)]$, meaning the baseline is zero in expectation. However, this derivation only holds for odd functions! Unfortunately, the second-order score function $\frac{\nabla_N^{(2)} q_\alpha(\mathbf{z}_i)}{q_\alpha(\mathbf{z}_i)}$ is an even function since the derivative of an odd function is always an even function. Therefore, the ARM estimator will only be unbiased for first-order gradient estimation.

### D.2.6 GO Gradient

The GO gradient estimator [7] is a method that uses the CDF of the distribution to derive the gradient. For continuous distributions, it reduces to implicit reparameterization gradients which can be implemented through transforming the computation graph, like other reparameterization methods. For $m$ independent discrete distributions of $d$ categories, the first-order gradient is given as:

$$\mathbb{E}_{p(\mathbf{x}_i|\mathbf{x}_{\leq i})}\Big[\sum_{j=1}^m (f(\mathbf{x}_{\leq i}) - f(\mathbf{x}_{\leq i \setminus \mathbf{x}_{i,j}}, \mathbf{x}_{i,j} + 1))\frac{\nabla_N \sum_{k=1}^{\mathbf{x}_{i,j}} p_j(k|\mathbf{x}_{<i})}{p_j(\mathbf{x}_{i,j}|\mathbf{x}_{<i})}\Big]$$

Note that if $\mathbf{x}_{i,j} = d$, then the estimator evaluates to zero since $\nabla_N \sum_{k=1}^d p_j(k|x_{<i}) = 0$.

We derive the *Storchastic* implementation by treating the GO estimator as a control variate of the single-sample score function. To find this control variate, we subtract the score function from this estimator, that is, we subtract $f(\mathbf{x}_{\leq i})\nabla_N \log p(\mathbf{x}_i|\mathbf{x}_{<i}) = f(\mathbf{x}_{\leq i})\sum_{j=1}^m \nabla_N \log p(\mathbf{x}_{i,j}|\mathbf{x}_{<i}) = f(\mathbf{x}_{\leq i})\sum_{j=1}^m \frac{\nabla_N p(\mathbf{x}_{i,j}|\mathbf{x}_{<i})}{p(\mathbf{x}_{i,j}|\mathbf{x}_{<i})}$ where we use that each discrete distribution is independent. By unbiasedness of the GO gradient, the rest of the estimator is 0 in expectation, as we will show.

Define $f_{j,k} = f(\mathbf{x}_{\leq i \setminus \mathbf{x}_{i,j}}, \mathbf{x}_{i,j} = k)$, $p_{j,k} = p_j(k|\mathbf{x}_{<i})$ and $P_{j,k} = \sum_{k'=1}^k p_j(k'|\mathbf{x}_{<i})$. Then the GO control variate is:

$$a_i(\mathbf{x}_{\leq i}) = \sum_{j=1}^m I[\mathbf{x}_{i,j} < d]\Big(\bot\big(\frac{f_{j,\mathbf{x}_{i,j}} - f_{j,\mathbf{x}_{i,j}+1}}{p_{j,\mathbf{x}_{i,j}}}\big)(\boxdot(P_{j,\mathbf{x}_{i,j}}) - 1)\Big) - \bot(f_{j,d})(\boxdot(\log p_{j,d}) - 1)$$

The first line will evaluate to the GO gradient estimator when differentiated, and the second to the single-sample score function gradient estimator.

Note that this gives a general formula for implementing any unbiased estimator into *Storchastic*: Use it as a control variate with the score function subtracted to ensure interoperability with other estimators in the stochastic computation graph.

### D.3 SPSA

Simultaneous perturbation stochastic approximation (SPSA) [46] is a gradient estimation method based on finite difference estimation. It stochastically perturbs parameters and uses two functional evaluations to estimate the (possibly stochastic) gradient. Let $\boldsymbol{\theta}$ be the $d$-dimensional parameters of

the distribution $p_{\boldsymbol{\theta}}(\mathbf{x}_i|\mathbf{x}_{<i})$. SPSA samples $d$ times from the Rademacher distribution (a Bernoulli distribution with 0.5 probability for 1 and 0.5 probability for -1) to get a noise vector $\boldsymbol{\epsilon}$. We then get two new distributions: $\mathbf{x}_{i,1} \sim p_{\boldsymbol{\theta}+c\boldsymbol{\epsilon}}$ and $\mathbf{x}_{i,2} \sim p_{\boldsymbol{\theta}-c\boldsymbol{\epsilon}}$ where $c > 0$ is the perturbation size. The difference $\frac{f(\mathbf{x}_{i,1})-f(\mathbf{x}_{i,2})}{2c\boldsymbol{\epsilon}}$ is then an estimate of the first-order gradient. Higher-order derivative estimation is also possible, but left for future work.

An easy way to implement SPSA in *Storchastic* is by using importance sampling (Appendix D.2.2). Assuming $p_{\boldsymbol{\theta}+c\boldsymbol{\epsilon}}$ and $p_{\boldsymbol{\theta}-c\boldsymbol{\epsilon}}$ have the same support as $p$, we can set the weighting function to $\perp\left(\frac{p(\mathbf{x}_{i,1}|\mathbf{x}_{<i})}{p_{\boldsymbol{\theta}+c\boldsymbol{\epsilon}}(\mathbf{x}_{i,1}|\mathbf{x}_{<i})}\right)$ for the first sample, and $\perp\left(\frac{p(\mathbf{x}_{i,2}|\mathbf{x}_{<i})}{p_{\boldsymbol{\theta}-c\boldsymbol{\epsilon}}(\mathbf{x}_{i,2}|\mathbf{x}_{<i})}\right)$ for the second sample.

To ensure the gradients distribute over the parameters, we define the gradient function as $\boldsymbol{\theta}\perp\left(\frac{p_{\boldsymbol{\theta}+c\boldsymbol{\epsilon}}(\mathbf{x}_{i,1}|\mathbf{x}_{<i})}{2c\boldsymbol{\epsilon}p(\mathbf{x}_{i,1}|\mathbf{x}_{<i})}\right)$ for the first sample and $-\boldsymbol{\theta}\perp\left(\frac{p_{\boldsymbol{\theta}+c\boldsymbol{\epsilon}}(\mathbf{x}_{i,2}|\mathbf{x}_{<i})}{2c\boldsymbol{\epsilon}p(\mathbf{x}_{i,2}|\mathbf{x}_{<i})}\right)$ for the second sample. This cancels out the weighting function, resulting in the SPSA estimator.

### D.4 Measure Valued Derivatives

*Storchastic* allows for implementing Measure Valued Derivatives (MVD) [18], however, it is only unbiased for first-order differentiation and cannot easily be extended to higher-order differentiation. The implementation is similar to SPSA, but with some nuances. We will give a simple overview for how to implement this method in *Storchastic*, and leave multivariate distributions and higher-order differentiation to future work.

First, define the weak derivative for parameter $\theta$ of $p$ as the triple $(c_\theta, p^+, p^-)$ by decomposing $p(\mathbf{x}_i|\mathbf{x}_{<i})$ into the positive and negative parts $p^+(\mathbf{x}_i^+)$ and $p^-(\mathbf{x}_i^-)$, and let $c_\theta$ be a constant. For examples on how to perform this decomposition, see for example [37]. To implement MVDs in *Storchastic*, we use the samples from $p^+$ and $p^-$, and, similar to SPSA, treat them as importance samples (Appendix D.2.2) for the zeroth order evaluation.

That is, the proposal distribution is defined over tuples $\mathcal{X}_i = (\mathbf{x}_i^+, \mathbf{x}_i^-)$ such that $q(\mathcal{X}_i|\mathbf{x}_{<i}) = p^+(\mathbf{x}_i^+)p^-(\mathbf{x}_i^-)$. The weighting function can be derived depending on the support of the positive and negative parts of the weak derivative. For weak derivatives for which the positive and negative part both cover an equal proportion of the distribution $p(\mathbf{x}_i|\mathbf{x}_{<i})$, the weighting function can be found using importance sampling by $\perp\left(\frac{p(\mathbf{x}_i^+|\mathbf{x}_{<i})}{2p^+(\mathbf{x}_i^+)}\right)$ for samples from the positive part, and $\perp\left(\frac{p(\mathbf{x}_i^-|\mathbf{x}_{<i})}{2p^-(\mathbf{x}_i^-)}\right)$ for samples from the negative part. This gives unbiased zeroth order estimation by using importance sampling.

We then set $a_i(\mathbf{x}_{<i}, \mathcal{X}_i) = 0$ and use the following gradient function: $l_i(\mathbf{x}_i) = \theta \cdot \perp(c_\theta \frac{2p^+(\mathbf{x}_i^+)}{p(\mathbf{x}_i^+|\mathbf{x}_{<i})})$ for positive samples and $l_i(\mathbf{x}_i) = -\theta \cdot \perp(c_\theta \frac{2p^-(\mathbf{x}_i^-)}{p(\mathbf{x}_i^-|\mathbf{x}_{<i})})$ for negative samples. This will compensate for the weighting function by ensuring the importance weights are not applied over the gradient estimates. For the first-order gradient, this results in the MVD $\nabla_N \theta \perp(c_\theta)(f(\mathbf{x}_i^+) - f(\mathbf{x}_i^-))$.

For other distributions for which $p^+$ and $p^-$ do not cover an equal proportion of $p$, more specific implementations have to be derived. For example, for the Poisson distribution one can implement its MVD by noting that $p^+$ has the same support as $p$. Then, we can use one sample from $p^+$ using the importance sampling estimator using score function (Appendix D.2.2), and use a trick similar to the GO gradient by defining a control variate that subtracts the score function and adds the MVD, which is allowed since the MVD and score function are both unbiased estimators.

## E  Discrete VAE Case Study Experiments

We report test runs on MNIST [26] generative modeling using discrete VAEs in Table 1. We use Storchastic to run 100 epochs on both a latent space of 20 Bernoulli random variables and 20 Categorical random variables of 10 dimensions, and report training and test ELBOs. We run these on the gradient estimation methods currently implemented in the PyTorch library.

Although results reported are worse than similar previous experiments, we note that we only run 100 epochs (900 epochs in [25]) and we do not tune the methods. However, the results reflect the order expected from [25], where score function with leave-one-out baseline also performed best,

| | $2^{20}$ Bernoulli VAE | | $10^{20}$ Discrete VAE | |
| --- | --- | --- | --- | --- |
| | Train ELBO | Validation ELBO | Train ELBO | Validation ELBO |
| Score@1 | 191.3 | 191.9 | 206.3 | 206.7 |
| ScoreLOO@5 [22] | 110.8 | 110.4 | 111.2 | 110.4 |
| REBAR@1 [47] | 220.0 | 1000 | 155.6 | 154.9 |
| RELAX@1 [16] | 210.6 | 205.9 | 202.5 | 201.7 |
| Unordered set@5 [25] | 117.1 | 138.4 | 115.4 | 117.2 |
| Gumbel@1 [19, 31] | 107.0 | 106.6 | 92.9 | 92.6 |
| GumbelST@1 [19] | 113.0 | 112.9 | 98.3 | 98.0 |
| ARM@1 [51] | 131.3 | 130.8 | | |
| DisARM@1[9] | 125.1 | 124.3 | | |

Table 1: Test runs on MNIST VAE generative modeling. We report the lowest train and validation ELBO over 100 epochs. The number after the '@' symbol denotes the amount of samples used to compute the estimator. We note that the ARM and DiSARM methods are specific for binary random variables, and do not evaluate it in the $10^{20}$ discrete VAE.

closely followed by the Unordered set estimator. Furthermore, the Gumbel softmax [19, 31] still outperforms the other score-function based estimators, although the results in [25] suggest that with more epochs and better tuning, better ELBO than reported here can be achieved.

These results are purely presented as a demonstration of the flexbility of the Storchastic library: Only a single line of code is changed to be able to compare the different estimators! A more thorough and fair comparison, also in different settings, is left for future work.