# OpenReview forum: "Storchastic: A Framework for General Stochastic Automatic Differentiation"
_NeurIPS.cc/2021/Conference — NeurIPS 2021 Poster_

### Official Review · Reviewer_3poh · 2021-07-11

**Rating:** 6
**Confidence:** 3

**Summary:**

This paper introduces a framework for stochastic gradient estimation that is able to capture many existing gradient estimation techniques, and allows a modeler to interchange different gradient estimation techniques in their computation graph. The paper also provides a mathematical framework to prove unbiasedness of any-order derivatives for common gradient estimation techniques using their surrogate loss.

The paper introduces new notation and a formalization of forward mode AD and gradient computation in stochastic graphs. Using this framework, they introduce an algorithm that, given a partition of stochastic nodes, and a specification of a gradient estimator for each stochastic node, returns a surrogate loss that can be arbitrarily differentiated to give an n-th order derivative of a cost wrt a parameter. Specification of a gradient estimator reduces to specifying 4 components: a gradient function, proposal distribution, weighting function, and control variate.

The paper then proves that many existing gradient estimation techniques fit in the framework, and for many of these techniques use the framework to prove unbiasedness of arbitrary derivatives.

**Limitations And Societal Impact:**

The most important missing piece I think is the lack of explicit case studies / experiments with the framework. Although I understand the framework provides a lot of flexibility to the modeler, is this flexibility useful? It would be very helpful for me to see some sort of demonstration of the framework.

How is the computation cost/overhead for the framework? You mention that the method is exponential in the number of stochastic nodes. I assume this is unavoidable if you want multiple samples per stochastic node, but it would be very helpful to understand if the framework adds any additional overhead over using one of the gradient estimators in a standard way. Having a case study section would also help answering this question.

**Main Review:**

I think the unification of several gradient estimation techniques into a framework is a nice contribution. The paper is well polished and clearly written, and the notation introduced seems to help with the proofs significantly.

Although similar to DiCE, it seems to extend it in a nontrivial way by incorporating many different gradient estimation functions, and bringing a more formal foundation to it.

The paper also introduces an implementation of the framework in PyTorch, but unfortunately does not include any case studies or experiments with the code that would help the reader understand whether the framework is particularly useful. I expand on this in the limitations section.

Given the limitations, I am borderline on the paper, but currently lean towards acceptance because I think the framework is a nice contribution.

**Time Spent Reviewing:**

4 hours

---

> ### Author Response · Authors · 2021-08-09
> **Response to reviewer 3poh**
>
> The reviewer appreciates the unification of gradient estimation methods into one framework and the formal foundation of the paper, and believes the paper is polished and clearly written. However, the reviewer also mentions the lack of an explicit case study. We will implement this suggestion by adding a code comparison of how to implement the leave-one-out baseline for a discrete variational autoencoder between Storchastic and vanilla Pytorch, and by running the implemented set of gradient estimators on the MNIST problem.
>
> > How is the computation cost/overhead for the framework? You mention that the method is exponential in the number of stochastic nodes. I assume this is unavoidable if you want multiple samples per stochastic node, but it would be very helpful to understand if the framework adds any additional overhead over using one of the gradient estimators in a standard way. Having a case study section would also help answering this question.
>
> We discuss questions related to the fact that it is exponential in the number of stochastic nodes in our response to R1's question, which we copy in the paragraphs below. As to the second, the framework itself adds no direct complexity compared to using the gradient estimators regularly, as it mainly works by reordering the way the computations are set up. However, the PyTorch implementation will have some overhead as we have to do some bookkeeping and transposing of tensors to automatically perform the parallelization. We will add running time comparisons with a vanilla PyTorch implementation to the case study.
>
> **Copied answer to R1 about exponential in number of samples**
>
> To reduce the overhead from using many samples, one can choose to use multiple samples only for a couple of partitions that require heavy variance reduction, while only using a single score function evaluation at other partitions. Furthermore, you could dynamically/randomly choose certain partitions to sample multiple times from to ensure variance reduction can be used on a limited budget. Such a trick is used in ARM (section 3.1, [1]) to reduce the $O(2^k)$ complexity in stochastic binary networks to $O(2k)$ complexity. This is possible here because the weights of the i'th layer only influence the i'th stochastic node, in general SCG's this is not possible.
>
> Another practical scenario is when using estimators like MAPO, Unordered Set Estimator and Rao-Blackwellized estimator, which have a fixed budget over a single partition of stochastic nodes.
>
> [1] Yin, Mingzhang, and Mingyuan Zhou. "ARM: Augment-REINFORCE-merge gradient for stochastic binary networks." arXiv preprint arXiv:1807.11143 (2018).
>
> > Although I understand the framework provides a lot of flexibility to the modeler, is this flexibility useful?
>
> Copied partially from our response to R1:
>
> A motivating example might be training a generative machine translation model with discrete latent space directly on the BLEU score. The gradient estimator of the latent space could be ARM-based (e.g., the new ARMS [1]), while sampling sequences to estimate gradients over the BLEU score could be done using, e.g., MAPO as BLEU is deterministic and MAPO is constant in sequences. We believe creating an experiment like this to be out of scope for the current paper, as it encompasses comparing empirically what estimators perform best in what settings.
>
> Furthermore, we believe our framework is also useful for less complex models. It can be used to develop new gradient estimation methods and it generalizes them to higher-order differentiation, which is a fairly common use case. Finally, by being able to swap-in different estimators by changing a single line of code, it will be much easier for practitioners to find the right estimator for their problem.
>
> [1] Dimitriev, Alek, and Mingyuan Zhou. "ARMS: Antithetic-REINFORCE-Multi-Sample Gradient for Binary Variables." arXiv preprint arXiv:2105.14141 (2021).

---

> > ### Comment · Reviewer_3poh · 2021-08-23
> > **Response to author response**
> >
> > Thank you for the reply, and agreeing to include a case study in the paper. The motivating example is also helpful to a certain extent.
> >
> > Although my instinct is still that this is a good paper and I am leaning towards acceptance even in its current form, I will keep my score at 6 and remain lower confidence. Seeing a case study would be very helpful for me as I am not an expert in stochastic gradient estimation and its practical usage. I think adding the case study to the paper really would improve the readability, motivation, and inspire people to use the framework.

---

### Official Review · Reviewer_VArr · 2021-07-16

**Rating:** 8
**Confidence:** 3

**Summary:**

Storchastic presents a general formulation for surrogate losses to be differentiated at any order to evaluate a large array of gradient estimation techniques for expectations of stochastic computational graphs and implements this framework in PyTorch.

The MagicBox operator idea of the DiCE estimator can be used to make the score function gradient estimator evaluatable by AD methods. The Storchastic framework utilizes this idea for a surrogate loss that works for random variables conditioned on other random variable in the computational graph. The required sample size grows exponentially in the depth of the conditional nesting. In the PyTorch implementation the nested sampling is implemented by extra dimensions in the tensors which allows for good parallelization.

In Storchastic each random variable can use a suitable individual variance reduction technique, such as proposal distribution with weighting or control variates (or the reparameterization trick). The paper demonstrates that the proposed surrogate loss can be used to implement various well-established techniques from the literature.

The paper shows that under certain reasonable local conditions the obtained estimators are unbiased under any order of differentiation. These conditions are as follows:
- for all orders of differentiation the surrogate should be unbiased for any deterministic function under the proposal distribution + weighting
- for all orders of differentiation the control variate should be unbiased under the proposal distribution + weighting
- the expectation of the weighting function needs to be zero when differentiated
- the proposal distribution should be id under forward mode evaluation

The paper also shows how control variates can be appropriately pulled into the surrogate loss formulation for nested conditional random variables using the MagicBox trick and provide variance reduction at any order of differentiation.


**Limitations And Societal Impact:**

Yes

**Main Review:**

The paper gives a succinct definition of stochastic computational graphs and the essential problem of estimating gradients (and higher order derivatives) of expectations. The definitions, problem statements, assumption and introduced formalisms are stated clearly and explained well.

The general formulation encapsulating existing gradient estimators and making the connection to modern frameworks for DNN training is an original contribution that has the potential to be of significant use for practitioners that are working on complex stochastic models (such as deep discrete variational auto encoders etc.).

The combination and generalization of the various different approaches to gradient estimation approaches in a general surrogate loss formalism as presented may also be helpful to researchers trying to derive gradient estimators even when they do not end up using the PyTorch implementation. Nonetheless, I think the paper could be strengthened by including a brief section about the implementation interface so that readers of the paper can already have an idea of how they would interact with the library.

I recommend to accept the paper because it presents a high quality clear methodology of how to elegantly implement state-of-the-art gradient estimators using AD methods.

Nit:
- The sum in Algorithm 1 reads like "s times u times m" (the italic letters look like individual variable names), maybe you could change the typesetting here

**Time Spent Reviewing:**

2

---

> ### Author Response · Authors · 2021-08-09
> **Response to reviewer VArr**
>
> The reviewer mentions the clarity and succinctness of our definition and formalization. They appreciate the originality and practicality of the contribution, and believes it can be particularly significant for practitioners in complex stochastic models. Finally, they mention the framework can be useful to develop new gradient estimators.
>
> Moreover, they suggest strengthening the paper by including a section on the implementation interface. We choose to adopt this suggestion by presenting code snippets comparing Storchastic and vanilla PyTorch for implementing the leave-one-out-baseline on discrete variational autoencoders.
>
> > The sum in Algorithm 1 reads like "s times u times m" (the italic letters look like individual variable names), maybe you could change the typesetting here
>
> Good point, thank you. We will do that!

---

### Official Review · Reviewer_6cXR · 2021-07-19

**Rating:** 6
**Confidence:** 3

**Summary:**

This paper introduces a framework for automatic differentiation for stochastic computation graphs. A major step in stochastic computation graph is the sampling step and this can happen when modelers deal with expectation terms in models such as reinforcement learning or variational inference. This framework provides an unbiased estimator for any order gradient and also generalizes the variance reduction technique for those estimates.  For gradient estimation, it needs a 4-tuple including proposal distribution-to sample from-, weighting function-to weight samples-, gradient function- unbiased gradient estimator- and control variate -to reduce variance-. Based on this 4-tuple it defines a surrogate loss to train the parameters of the graph. On the theoretical side, it shows that its estimator is unbiased and on the practical side this framework is implemented in PyTorch.

**Limitations And Societal Impact:**

Yes

**Main Review:**

This work considers an important practical problem that modelers deal with when implementing stochastic models. However, their work is mainly based on previous work such as [13].

The major comment I have with regard to this paper is about its clarity. The significance of this problem is not well described for an unfamiliar reader. For example, they could start with a VI model and explain why this is hard to implement in PyTorch. Furthermore, they use the magic box operator from previous work but they don’t explain clearly what it does and why it is important in SCG models. Another part is the section where they explain their surrogate loss. No intuition is given to the reader about how they come up with this loss function.
Also, it would be nice to see more detail in the mathematical form about the rule of control variate in reducing the variance in example 3.2.5.

My questions:

1- Is the proposal distribution fixed through the training or can it be changed?

2-   How can the modeler determine the cardinality of samples for each stochastic node in different partitions? Is it fixed or dynamic?


3-  Does this framework allow arbitrary weighting for different cost nodes? Can we train these weights as well?


4- Not including the reparameterization trick is a limitation. How much change does your framework need to include this trick?


**Time Spent Reviewing:**

4

---

> ### Author Response · Authors · 2021-08-09
> **Response to reviewer 6cXR**
>
> We thank the reviewer for their work and comments. While they believe the paper considers an important practical problem, they have some comments about the clarity of the paper and a couple of interesting questions. We hope to address both.
>
> To improve the clarity and motivation, they suggest
> >  they could start with a VI model and explain why this is hard to implement in PyTorch.
>
> We believe this is a strong idea, which works well together with an R1's suggestion to compare code snippets. We will adopt this in the final version by considering the discrete variational autoencoder, with gradient estimation using the score function with leave-one-out baseline, as an example. Furthermore, we will introduce this example early on in the paper to ease the reader into the motivation for the framework.
>
> > they use the magic box operator from previous work but they don’t explain clearly what it does and why it is important in SCG models
>
> We tried to introduce the MagicBox succinctly in Section 2.3 and Definition 1, with additional insights in Appendix A. We will move some insights from the appendix to the main text to introduce it with more detail.
>
> > Another part is the section where they explain their surrogate loss. No intuition is given to the reader about how they come up with this loss function.
>
> This is a fair point. The reason we do not have a 'linear' reasoning to build towards this loss function is because we iterated on the framework repeatedly to fulfil all requirements listed in Section 3.1. We attempted to provide intuition in how we got to the final result by linking individual components of the framework to these requirements. We will give this another read to improve the clarity.
>
> > it would be nice to see more detail in the mathematical form about the rule of control variate in reducing the variance in example 3.2.5.
>
> We will add a comment showing how the baseline would have been implemented without Storchastic to better facilitate a mathematical insight in its role.
>
> The 4 questions the reviewer had:
>
> > Is the proposal distribution fixed through the training or can it be changed?
>
> Yes, it can be changed. The conditions in the proof only require that the components are unbiased in a single iteration. Each iteration can be a different (unbiased) gradient estimator and therefore also a different proposal distribution.
>
> > How can the modeler determine the cardinality of samples for each stochastic node in different partitions? Is it fixed or dynamic?
>
> As a corollary of the last question, yes, you can dynamically select how many samples to take at a stochastic node. This can be used, for instance, to dynamically allocate extra computation to some stochastic nodes while keeping the whole system tractable, as explained in our answer to R1 about exponential time.
>
> > Does this framework allow arbitrary weighting for different cost nodes? Can we train these weights as well?
>
> Yes, cost nodes can be arbitrarily weighted by simply multiplying the output of a cost node with a weight and making this the new weight. "Training" these weights would be possible, but directly minimizing a cost function with weights would just set all weights to 0.
>
> > Not including the reparameterization trick is a limitation. How much change does your framework need to include this trick?
>
> This is a good question! We can add reparameterization by introducing a new component, let's call it 'transformation function’, which transforms the sampled values. For everything but reparameterization, this would just be the identity function. We can then implement reparameterization by using a proposal distribution that's simply random noise, then using the transformation function to transform noise into a proper sample.
> We do not include this into our framework to decrease the complexity, and because reparameterization is simple to implement in AD libraries given a resample implementation.

---

### Official Review · Reviewer_xpUA · 2021-07-21

**Rating:** 6
**Confidence:** 2

**Summary:**

The paper presents "storchastic", a framework which allows choice and combination of a variety of methods for (potentially higher-order) gradient estimation.  The paper shows how many gradient estimators and control variates can be reconstructed within this framework, and proves that many of the resulting (potentially higher-order) gradient estimators are unbiased given some conditions, and show that control variates can reduce the variance of even higher-order gradient estimates.

**Limitations And Societal Impact:**

Yes

**Main Review:**

I am not very confident in my assessment of this paper.

The paper describes what sounds like a very useful library, allowing a choice of gradient estimators (proposal distribution, weight function, gradient function, control variate) and facilitating higher-order gradient estimation. The proofs of conditions for unbiased higher-order gradient estimation and any-order variance reduction seem useful and important.

With the exception of these two proofs, most of the contribution is in the framework / library.  The paper describes the desiderata and approach, but there is not really any qualitative or quantitative evaluation of the library.  It is much more difficult to come up with evaluation/comparison for a paper like this than it is for a method or model which can be measured on some benchmark.

I lean slightly to acceptance as the description of the framework makes the library sound like a useful tool, and as I think we should encourage development of such open-source tools and libraries.  The two proofs are also useful contributions.

However I think the paper could be significantly improved if the library/framework could  be demonstrated and could somehow be compared with existing libraries. For example, code snippets could show the ease of implementing different gradient estimators with Storchastic vs in vanilla Pytorch or in Pyro. Experiments showing the efficacy of control variates for higher-order gradient estimation, or showing the efficacy of using different gradient estimators at different stochastic nodes, would also support the premise that this library's functionality is an important improvement.

The computation being exponential in the size of the sampled sets of values X seems like a major limitation. Do prior libraries also have this limitation? Is it possible to overcome this, at the cost of variance, while still being able to reduce variance with increased sample size? I don't find "this can be parallelized" convincing as an argument for exponential computation time not mattering.

Nits -
Don't capitalize "Deep Learning", "Reinforcement Learning" or "Variational Inference", only capitalize proper nouns.
Is "Stochastic automatic differentiation" an accepted term? To me, it seems a bit ambiguous . Stochastic gradient descent, using  AD to compute gradients, sounds like it would fit "stochastic automatic differentiation", but it doesn't fit the definition from this paper, which involves gradient estimation in computation graphs where the parameters control the sampling distributions.
I also think it would be possible to have a debate over whether this is truly still automatic differentiation, given the user needs to specify what estimation methods to use at each node (but this is minor / particularly pedantic).

**Time Spent Reviewing:**

3

---

> ### Author Response · Authors · 2021-08-09
> **Response to reviewer xpUA**
>
> We want to thank the reviewer for the useful comments and will use their suggestions to improve our paper. We appreciate that the reviewer believes that both our library and the presented proofs are useful, and that they believe open-source software should be encouraged. However, the reviewer mentions the paper can be strengthened by adding a comparison to the paper:
>
> > I think the paper could be significantly improved if the library/framework could be demonstrated and could somehow be compared with existing libraries. For example, code snippets could show the ease of implementing different gradient estimators with Storchastic vs in vanilla Pytorch or in Pyro.
>
> We agree with this suggestion, and will add a code comparison of how to implement the leave-one-out baseline for a discrete variational autoencoder between Storchastic and vanilla Pytorch for the camera ready.
>
> We also considered comparing with Pyro, however, we believe this comparison to be confusing. This is because
> 1. Pyro is a higher-level library than Storchastic, serving a different end-user need,
> 2. it doesn't have a clear interface for implementing additional gradient estimators and
> 3. Pyro could use Storchastic for its stochastic variational inference implementation (it currently uses DiCE, which we generalize).
>
> > Experiments showing the efficacy of control variates for higher-order gradient estimation, or showing the efficacy of using different gradient estimators at different stochastic nodes, would also support the premise that this library's functionality is an important improvement.
>
> We agree this would further strengthen it. We will improve our motivation by referring to the experiments in [3], which show empirically that higher-order variance reduction indeed improves convergence in a multi-agent setting that requires higher-order estimation.
> An experiment that shows the efficacy of using different gradient estimators at different stochastic nodes would also be valuable. However, we were not able to come up with a small toy problem where this is obvious.
>
> A motivating example might be training a generative machine translation model with discrete latent space directly on the BLEU score. The gradient estimator of the latent space could be ARM-based (e.g., the new ARMS [4]), while sampling sequences to estimate gradients over the BLEU score could be done using, e.g., MAPO as BLEU is deterministic and MAPO is constant in sequences. We believe this experiment to be out of scope for the current paper, as it essentially encompasses thoroughly comparing empirically what estimators perform best in what settings.
>
> > The computation being exponential in the size of the sampled sets of values X seems like a major limitation. Do prior libraries also have this limitation? Is it possible to overcome this, at the cost of variance, while still being able to reduce variance with increased sample size?
>
> First, we want to clarify that the paper is currently not clear on this exponential time dependency. It is not exponential in the size of the sampled set of values, but in the amount of partitions. More formally, the complexity would be $O(|X_j|^k)$, where $|X_j|$ is the cardinality of the largest set of sampled values, and $k$ is the amount of partitions of stochastic nodes. Prior libraries (e.g., Pyro) also have this limitation. The exponential time in the amount of partitions is an artefact of the gradient estimation method chosen. Storchastic simply aims to faithfully implement those. We will correct and clarify this in the camera ready.
>
> To reduce the overhead from using many samples, one can choose to use multiple samples only for a couple of partitions that require heavy variance reduction, while only using a single score function evaluation at other partitions. Furthermore, you could dynamically/randomly choose certain partitions to sample multiple times from to ensure variance reduction can be used on a limited budget. Such a trick is used in ARM (section 3.1, [2]) to reduce the $O(2^k)$ complexity in stochastic binary networks to $O(2k)$ complexity. This is possible here because the weights of the i'th layer only influence the i'th stochastic node, in general SCG's this is not possible.
>
> Another practical scenario is when using estimators like MAPO, Unordered Set Estimator and Rao-Blackwellized estimator, which have a fixed budget over a single partition of stochastic nodes.
>
> > Is "Stochastic automatic differentiation" an accepted term?
>
> Admittedly, not really. However, SGD is actually an instance of stochastic automatic differentiation (consider SCG with all stochastic nodes external/as inputs to the graph). We use the term because 'gradient estimation in stochastic computation graphs' is very verbose.
>
> > I also think it would be possible to have a debate over whether this is truly still automatic differentiation, given the user needs to specify what estimation methods to use at each node
>
> That is fair. One could try to analyze the graph to give meaningful presets, to still make it automatic. We in fact do this in code to automatically switch between reparameterization and the score function.
>
> Finally, we will make sure to properly capitalize nouns in the camera ready as suggested.
>
>
> - [1] Obermeyer, Fritz, et al. "Tensor variable elimination for plated factor graphs." International Conference on Machine Learning. PMLR, 2019.
> - [2] Yin, Mingzhang, and Mingyuan Zhou. "ARM: Augment-REINFORCE-merge gradient for stochastic binary networks." arXiv preprint arXiv:1807.11143 (2018).
> - [3] Mao, Jingkai, et al. "A baseline for any order gradient estimation in stochastic computation graphs." International Conference on Machine Learning. PMLR, 2019.
> - [4] Dimitriev, Alek, and Mingyuan Zhou. "ARMS: Antithetic-REINFORCE-Multi-Sample Gradient for Binary Variables." arXiv preprint arXiv:2105.14141 (2021).

---

### Author Response · Authors · 2021-08-09
**General response to the reviewers**

We would like to thank the reviewers for their work in reviewing our paper.
We are happy to see the reviewers agree with us about the importance of the problem (R2) and that they think the framework (R3, R4) and library (R1) are worthwhile contributions that could be of significant use to practitioners (R3). The reviewers in particular appreciate the flexibility of the framework (R1), that it facilitates higher-order gradient estimation (R1) and that it unifies gradient estimators (R4). The reviewers also mention the usefulness of the formal proofs (R1, R4).

However, a common point of feedback is that the paper could be strengthened by a case study of using Storchastic (R1, R3, R4). R1 suggests comparing how to implement gradient estimators using Storchastic compared to existing methods, while R3 suggests a brief section on the implementation interface of the library. Although we are not allowed to submit a revision during the reviewing period, we will improve the camera ready by following these suggestions. We will do so by including code snippets comparing how to implement a discrete variational autoencoder with the leave-one-out baseline both in vanilla PyTorch and using Storchastic. We will introduce this problem early on in the paper to improve the clarity and motivate the significance of gradient estimation in SCGs, as suggested by R2. Furthermore, we will add a small experiment where we run all implemented gradient estimators on the discrete VAE with MNIST, a common benchmark in the literature.


For clarity, we use
- R1 = xpUA
- R2 = 6cXR
- R3 = VArr
- R4 = 3poh

---

### Decision · Program_Chairs · 2021-09-27

**Decision:**

Accept (Poster)

**Comment:**

This paper introduces a new framework for stochastic gradient estimation that encompasses various existing gradient estimation techniques. The unification of several gradient estimation techniques into a single framework is recognized by the reviewers as a somewhat significant contribution. The reviewers did not raise any severe concern so overall, I think this is a worthwile contribution as a poster.
One reviewer asked for a case study which I would like to authors to include in a revision given that they promised to do so.